

# Gravitational wave induced baryon acoustic oscillations

**Christian Döring[1*], Salvador C. Chuliá[2†], Manfred Lindner[1‡],**
**Bjoern M. Schaefer[3∘] and Matthias Bartelmann[4§]**

**1** Max-Planck-Institut für Kernphysik, Saupfercheckweg 1, 69117 Heidelberg, Germany
**2** AHEP Group, Institut de Física Corpuscular – C.S.I.C./Universitat de València ,
Parc Científic de Paterna. C/ Catedrático José Beltrán, 2 E-46980 Paterna (Valencia), SPAIN
**3** Zentrum für Astronomie der Universität Heidelberg, Astronomisches Rechen-Institut,
Philosophenweg 12, 69120 Heidelberg, Germany
**4** Institut für Theoretische Physik, Heidelberg University, Philosophenweg 16 ,
69120 Heidelberg, Germany

⋆ cdoering@mpi-hd.mpg.de, † salcen@ific.uv.es, ‡ lindner@mpi-hd.mpg.de ,
∘ bjoern.malte.schaefer@uni-heidelberg, § bartelmann@uni-heidelberg.de

## Abstract

We study the impact of gravitational waves originating from a first order phase transition on structure formation. To do so, we perform a second order perturbation analysis in the $1 + 3$ covariant framework and derive a wave equation in which second order, adiabatic density perturbations of the photon-baryon fluid are sourced by the gravitational wave energy density during radiation domination and on sub-horizon scales. The scale on which such waves affect the energy density perturbation spectrum is found to be proportional to the horizon size at the time of the phase transition times its inverse duration. Consequently, structure of the size of galaxies and bigger can only be affected in this way by relatively late phase transitions at $\geq 10^6$ s. Using cosmic variance as a bound we derive limits on the strength $\alpha$ and the relative duration $(\beta/H_*)^{-1}$ of phase transitions as functions of the time of their occurrence which results in a new exclusion region for the energy density in gravitational waves today. We find that the cosmic variance bound forbids only relative long lasting phase transitions, e.g. $\beta/H_* \lesssim 6.8$ for $t_* \approx 5 \times 10^{11}$ s, which exhibit a substantial amount of supercooling $\alpha > 20$ to affect the matter power spectrum.

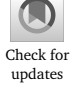

# 1   Introduction

With the first ever-measurement of a gravitational wave (GW) signal in 2016 from a black hole binary merger by the LIGO-Virgo collaboration [1] (now LIGO-Virgo-KAGRA), a new window of probing the universe has been opened. While this technique probes so far mostly astrophysical processes, future experiments like the space interferometer LISA [2] have the potential to explore also cosmological sources like first order phase transitions (FPT) in the early universe. In particle physics these phase transitions occur when the dropping temperature of the universe causes the vacuum expectation value (VEV) of a field to change discontinuously. If the field is hindered for a while from adapting to the new VEV by a barrier in its potential then bubbles enclosing the new VEV form, expand and eventually fill the universe with the new VEV. While such FPTs can produce GWs via the dynamics of the bubbles like collisions, soundwave formation and magneto-hydrodynamical effects, second order phase transitions and cross overs are not expected to produce substantial amounts of GWs, essentially because they lack the mechanism of vacuum bubble formation. The latter applies to the standard model of particles physics (SM), well described by the symmetry group $SU(3)_{QCD} \times SU(2)_L \times U(1)_Y$. It undergoes a cross over phase transition during the electro-weak symmetry breaking $SU(2)_L \times U(1)_Y \rightarrow U(1)_{QED}$ when the Higgs boson acquires a non-zero VEV [3, 4] and hence no GW signal is expected. The SM has, however, various problems, motivating for new physics beyond the SM (BSM). Many alternative models which incorporate new symmetries and particles allow for FPTs. The observation of GWs has therefore triggered many studies of FPTs in BSM models [5–12] where often GWs are expected to be seen in future GW experiments. For reviews see e.g. [13–17]. Future GW experiments can therefore valuably constrain BSM physics.

However, adding a FPT to the history of the universe might also affect other cosmological pro-

cesses such as formation of structure. This potential consequence is studied in this work. In the standard model of cosmology, linear density perturbations develop from inflation and seed over- and under densities in the various fluid components of the early cosmological medium. They propagate through the universe and undergo, depending on their scale, various changes caused by physical processes like the decoupling of a fluid component until they eventually form the structure we observe today. FPTs and the emerging GWs might influence this evolution depending on strength and duration. GWs are linear tensor perturbations of the metric sourced by an anisotropic stress distortion in the fluid while density perturbations of the fluid are scalar perturbations of the metric. At linear order in perturbation theory, they do not couple, but they can interact at second order and source second order density perturbations. Hence, we have to perform a second order expansion in order to capture effects that strong GW events may have. Typically, phase transitions are expected to occur while the universe is dominated by radiation and on sub-horizon scales. Consequently, potential effects on density perturbations are tied to the scale and thus the time of the transition. We shall work within the $1+3$ covariant approach to gravity [18–23] in which an exact, non linear equation of density perturbations is given.

Imprints of phase transitions in the matter power spectrum[1] have been of interest in the past, [25,26]. In contrast to our work these papers focus on the QCD phase transition during which they predicted a significant drop in the sound speed. This in turn affects the preexisting linear density perturbations and induces large peaks in the Harrison-Zel'dovich spectrum. Similar effects could happen in BSM transitions if the sound speed drops significantly which could be possible for theories with massive fermions or weakly interacting scalars but is not expected e.g. in simple scalar extensions of the SM [27].

This work is organized as follows. In Sec. 2 we investigate second order density perturbations and their coupling to GWs. In Sec. 3 we then summarize the physics of GWs from FPTs and present our results in Sec. 4. Subsequently we discuss in which way and under which conditions the GWs from FPTs do or do not affect the matter power spectrum, but also debate the limitations of our approach. Finally we conclude and give an outlook in Sec. 5. Further leading material and many details can additionally be found in the attached appendix.

## 2    Second order density perturbations

Let us begin with the study of second order density perturbations and the search for an equation in which density perturbations are sourced by GWs. To do so, we use the 1+3 covariant approach to cosmological perturbation theory for which a pedagogical introduction can be found in Appendix A. In this formulation spacetime is decomposed into the direction of the four-velocity, $u_a$, along the world line of a fundamental comoving observer and in its orthogonal direction, $h_{ab}$, where Latin indices run from 0 to 3. The energy momentum tensor of the cosmic fluid is split according to this decomposition

$$T_{ab} = \rho u_a u_b + 2u_{(a}q_{b)} + ph_{ab} + \pi_{ab}, \tag{1}$$

where the individual components are $\rho := T^{ab}u_a u_b$ the *energy density*, $q_b := h_a^{\ b}T_{bc}u^c$ the *energy current density*, $p := T_{ab}h^{ab}/3$ the *pressure* and $\pi_{ab} := T_{\langle ab\rangle}$ the trace-free *anisotropic stress*. Geometric quantities emerge from the splitting of the covariant derivative of the four-velocity. This includes the *shear* tensor $\sigma_{ab} := D_{\langle b}u_{a\rangle}$, the antisymmetric (hence tracefree) *vorticity* tensor $\omega_{ab} := D_{[b}u_{a]}$, the *volume expansion* scalar $\Theta := D^a u_a$ and the four-*acceleration*

---

[1]Linear matter power spectrum and measurements can be found in [24].

$A_a = u^b \nabla_b u_a$ such that

$$\nabla_b u_a = \sigma_{ab} + \omega_{ab} + \frac{1}{3}\Theta h_{ab} - A_a u_b\,. \tag{2}$$

Then the comoving density gradient and the comoving volume gradient are given as

$$\Delta_a := \frac{a}{\rho}D_a\rho\,, \tag{3}$$

$$Z_a := aD_a\Theta\,, \tag{4}$$

where the first one involves the notion of density contrast. It can be shown [28] by taking into account the equations and constraints from the Bianchi identities that the projected comoving density gradient and the projected comoving volume gradient evolve according to the full non-linear equations

$$\begin{aligned}
\dot{\Delta}_{\langle a\rangle} = {} & \frac{p}{\rho}\Theta\Delta_a - \left(1 + \frac{p}{\rho}\right)Z_a + a\frac{\Theta}{\rho}\left(\dot{q}_{\langle a\rangle} + \frac{4}{3}\Theta q_a\right) - \frac{a}{\rho}D_a D^b q_b + a\frac{\Theta}{\rho}D^b\pi_{ab} \\
& - \left(\sigma^b{}_a + \omega^b{}_a\right)\Delta_b - \frac{a}{\rho}D_a\left(2A^b q_b + \sigma^{bc}\pi_{bc}\right) + a\frac{\Theta}{\rho}\left(\sigma_{ab} + \omega_{ab}\right)q^b + a\frac{\Theta}{\rho}\pi_{ab}A^b \\
& + \frac{1}{\rho}\left(D^b q_b + 2A^b q_b + \sigma^{bc}\pi_{bc}\right)\left(\Delta_a - aA_a\right)\,,
\end{aligned} \tag{5}$$

and

$$\begin{aligned}
\dot{Z}_{\langle a\rangle} = {} & -\frac{2}{3}\Theta Z_a - \frac{1}{2}\kappa\rho\Delta_a - \frac{3}{2}\kappa aD_a p - a\left[\frac{1}{3}\Theta^2 + \frac{1}{2}\kappa(\rho + 3p) - \Lambda\right]A_a + aD_a D^b A_b \\
& - \left(\sigma^b{}_a + \omega^b{}_a\right)Z_b - 2aD_a\left(\sigma^2 - \omega^2\right) + 2aA^b D_a A_b \\
& - a\left[2\left(\sigma^2 - \omega^2\right) - D^b A_b - A^b A_b\right]A_a\,.
\end{aligned} \tag{6}$$

Here $\sigma^2 := \frac{1}{2}\sigma_{ab}\sigma^{ab}$ and $\omega^2 := \frac{1}{2}\omega_{ab}\omega^{ab}$. In order to find an expression that describes the influence that gravitational waves could induce on density perturbations, we seek for a relation between the orthogonal projected gradient $\Delta_a$ and linear perturbations of the shear tensor $\sigma^{(1)}_{ab}$, since the latter one describes the effects of GWs in the 1+3 approach. This occurs for the first time at second perturbative order in the density gradient $\Delta^{(2)}_a$. Therefore, in the following we will use Eqs. (5) and (6) to derive a linear equation for the time evolution of the density contrast $\Delta^{(2)}_a$ with a non-zero linear shear contribution $\sigma^{(1)}_{ab}$.

To do so, we will choose a model for the cosmic fluid which will significantly simplify the non-linear equations. Then, we will resolve the remaining angular brackets in the indices and take the orthogonal projected gradient of the equations in order to obtain scalar equations. While terms that are at least of third order will be directly neglected during the calculation, the explicit expansion of the remaining quantities to second order is performed after obtaining the scalar equation. Finally we set the background cosmology to FLRW and specialize the result for a radiation dominated fluid. During the calculation we will set $\kappa = 8\pi G = 1$ and reintroduce the units at the end. For our fluid model we impose the requirements

**Assumption 1:** At the background level, the matter-energy density is described by a (single component) perfect fluid.

**Assumption 2:** To all orders, we assume a negligible contribution from vorticity $\omega_{ab} = 0 = \omega_a$, current density $q_a = 0$ and anisotropy $\pi_{ab} = 0$ in the fluid.

In this model the full non linear equations Eq. (5) and Eq. (6) reduce to

$$\dot{\Delta}_{\langle a \rangle} = \frac{p}{\rho}\Theta\Delta_a - \left(1 + \frac{p}{\rho}\right)Z_a - \sigma_{ab}\Delta^b, \tag{7}$$

with

$$\dot{Z}_{\langle a \rangle} = -\frac{2}{3}\Theta Z_a - \frac{1}{2}\rho\Delta_a - \frac{3}{2}aD_a p + a\dot{\Theta}A_a + aD_a D^b A_b - \sigma_{ab}Z^b - 2aD_a\sigma^2 + 2aA^b D_a A_b. \tag{8}$$

Additionally we fix the relation between pressure and energy density by imposing

**Assumption 3:** Perfect barotropic fluid: This implies $p = \omega\rho$ with constant $\omega$ and together with Eq. (A.21) and $q_a = \pi_{ab} = 0$ the acceleration is to all orders $A_a = \frac{-c_s^2\Delta_a}{a(1+\omega)}$ due to $D_a p = \frac{\rho}{a}c_s^2\Delta_a$. The perturbations are adiabatic. We also assume $D_a\omega = \dot{\omega} = D_a c_s^2 = \dot{c}_s^2 = 0$.

Our perturbation procedure will extend to second order and thus we can neglect terms that are at least of third order in advance. This is the case for the term $a\left(2\sigma^2 - A^b A_b\right)A_a \gtrsim \mathcal{O}(\epsilon^3)$ since the acceleration $A_a$ and the shear $\sigma_{ab}$ are zero at zero order. Applying the third assumption to Eq. (7) and Eq. (8) our equations reduce to

$$\dot{\Delta}_{\langle a \rangle} = \omega\Theta\Delta_a - (1+\omega)Z_a - \sigma_{ab}\Delta^b, \tag{9}$$

$$\dot{Z}_{\langle a \rangle} = -\frac{2}{3}\Theta Z_a - \left(1+3c_s^2\right)\frac{\rho}{2}\Delta_a - \frac{c_s^2}{1+\omega}\dot{\Theta}\Delta_a - \frac{c_s^2}{a(1+\omega)}D_a\Delta - \sigma_{ab}Z^b$$

$$- 2aD_a\sigma^2 + \frac{c_s^4}{a(1+\omega)^2}D_a\left(\Delta^b\Delta_b\right). \tag{10}$$

The next step is to deal with the projected time derivative of the density perturbation. We expand it by applying the inverse product rule

$$\dot{\Delta}_{\langle a \rangle} := h_a^{\ b}\dot{\Delta}_b = (h_a^{\ b}\Delta_b)\dot{} - \dot{h}_a^{\ b}\Delta_b, \tag{11}$$

but since $h_a^{\ b}D_b = D_a$ and $u^a\Delta_a = u^a h_a^{\ b}\nabla_b = 0$ we find for the two terms

$$h_a^{\ b}\Delta_b := h_a^{\ b}\frac{a}{\rho}D_b\rho = \frac{a}{\rho}D_a\rho = \Delta_a, \tag{12}$$

$$\dot{h}_a^{\ b}\Delta_b = (u_a A^b + u^b A_a)\Delta_b = u_a A^b\Delta_b. \tag{13}$$

We thus get

$$\dot{\Delta}_{\langle a \rangle} = \dot{\Delta}_a - u_a A^b\Delta_b, \tag{14}$$

which reflects the fact that the orthogonally projected time variation of density inhomogeneities is the same as the complete time derivative of the density perturbation minus the projection of the density perturbation on the flow lines. The four acceleration in turn can be expressed by the relation $A_a = -\frac{c_s^2}{a(1+\omega)}\Delta_a$ and we find

$$\dot{\Delta}_{\langle a \rangle} = \dot{\Delta}_a + u_a\frac{c_s^2}{a(1+\omega)}\Delta^b\Delta_b. \tag{15}$$

For the expansion gradient we repeat this calculation and find

$$\dot{Z}_{\langle a \rangle} = h_a^{\ b}\dot{Z}_b = \left(h_a^{\ b}Z_b\right)\dot{} - \dot{h}_a^{\ b}Z_b = \dot{Z}_a - u_a A^b Z_b, \tag{16}$$

for the same reason as for the density perturbations, $h_a^{\ b}Z_b = Z_a$ and $u^a Z_a = 0$. Plugging these identities into Eqs. (9) and (10) results in

$$\dot{\Delta}_a + u_a \frac{c_s^2}{a(1+\omega)}\Delta^b \Delta_b = \omega\Theta\Delta_a - (1+\omega)Z_a - \sigma_{ab}\Delta^b, \tag{17}$$

$$\dot{Z}_a + u_a \frac{c_s^2}{a(1+\omega)}\Delta^b Z_b = -\frac{2}{3}\Theta Z_a - \left(1+3c_s^2\right)\frac{\rho}{2}\Delta_a - \frac{c_s^2}{1+\omega}\dot{\Theta}\Delta_a - \frac{c_s^2}{a(1+\omega)}D_a\Delta$$

$$- \sigma_{ab}Z^b - 2aD_a\sigma^2 + \frac{c_s^4}{(1+\omega)^2 a}D_a\left(\Delta^a \Delta_a\right). \tag{18}$$

## 2.1 Taking the orthogonal projected gradient

We are interested in the density fluctuations described by the comoving divergence of the density perturbations $\Delta := aD^a\Delta_a$. Hence, we will take the divergence of Eqs. (17) and (18) which will yield a scalar equation.

**Comoving fractional density gradient:** Let us start with the divergence of the first term in Eq. (17) which gives according to [29]

$$aD^a\dot{\Delta}_a = ah^{ab}\nabla_b u^c\nabla_c\Delta_a + ah^{ab}u^c\nabla_b\nabla_c\Delta_a \tag{19}$$

$$= \dot{\Delta} + \sigma^{ab}\Delta_{\langle ab\rangle} - \omega^{ab}\Delta_{[ab]} + \frac{1}{3}a\Theta A^a\Delta_a - aA^a\dot{\Delta}_a - aq^a\Delta_a + a\left(\sigma^{ab} + \omega^{ab}\right)\Delta_a A_b.$$

Applying our assumptions to this equation sets $q_a = \omega_{ab} = 0$. Ignoring again terms vanishing at second order we find

$$aD^a\dot{\Delta}_a = \dot{\Delta} + \sigma^{ab}\Delta_{\langle ab\rangle} - \frac{c_s^2}{3(1+\omega)}\Theta\Delta^a\Delta_a + \frac{c_s^2}{1+\omega}\Delta^a\dot{\Delta}_a + \mathcal{O}(\epsilon^3)$$

$$= \dot{\Delta} + \sigma^{ab}\Delta_{\langle ab\rangle} + \frac{c_s^2}{1+\omega}\left[\frac{1}{2}\frac{d}{dt} - \frac{1}{3}\Theta\right]\Delta^a\Delta_a. \tag{20}$$

We have also replaced $A_a = -\frac{c_s^2}{a(1+\omega)}\Delta_a$. Using $u^aD_a = 0$ the second term in Eq. (17) becomes

$$aD^a u_a \frac{c_s^2}{a(1+\omega)}\Delta^b\Delta_b = \frac{c_s^2}{a(1+\omega)}\Delta^b\Delta_b\, aD^a u_a = \frac{c_s^2}{(1+\omega)}\Delta^b\Delta_b\Theta. \tag{21}$$

The first term on the right hand side of Eq. (17) becomes

$$aD^a\omega\Theta\Delta_a = \omega(Z^a\Delta_a + \Theta\Delta), \tag{22}$$

while the second term reads

$$-(1+\omega)aD^a Z_a = -(1+\omega)Z. \tag{23}$$

Moving on to the third term we note that in general the space-like constraint on the shear is not zero but rather given by Eq. (A.25). However, in our model the shear plays the role of GWs. Hence we only consider the transverse component of the shear, thus $D^b\sigma_{ab} = 0$ (following [28–30]). This implies for the last term in Eq. (17)

$$aD^a(-\sigma_{ab}\Delta^b) = -(a\underbrace{D^a\sigma_{ab}\Delta^b}_{=0} + a\,\sigma_{ab}D^a\Delta^b) \tag{24}$$

$$= -\sigma_{ab}\left(\frac{1}{2}\Delta h^{ba} + \Delta^{\langle ba\rangle} + \Delta^{[ba]}\right) \tag{25}$$

$$= -\sigma_{ab}\Delta^{\langle ab\rangle}, \tag{26}$$

where we have made use of the fact that the shear is also tracefree $\sigma_{ab}h^{ab} = \sigma^a{}_a = 0$ and that the complete contraction of an antisymmetric with a symmetric tensor vanishes. Altogether, the differential equation for density fluctuations becomes

$$\dot{\Delta} = \omega(Z^a\Delta_a + \Theta\Delta) - (1+\omega)Z - 2\sigma_{ab}\Delta^{\langle ab\rangle} - \frac{c_s^2}{1+\omega}\left(\frac{2}{3}\Theta + \frac{1}{2}\frac{d}{dt}\right)\Delta^a\Delta_a. \quad (27)$$

**Comoving volume expansion:** Now we move on to the differential equation for the volume expansion gradient $Z_a$ in Eq. (18). Starting on the left hand side analogously to $\dot{\Delta}$ we find for the first term

$$a D^a \dot{Z}_a = \dot{Z} + \sigma^{ab}Z_{\langle ab\rangle} + \frac{1}{3}a\Theta A^a Z_a - aA^a \dot{Z}_a$$

$$= \dot{Z} + \sigma^{ab}Z_{\langle ab\rangle} - \frac{c_s^2}{3(1+\omega)}\Theta\Delta^a Z_a + \frac{c_s^2}{1+\omega}\Delta^a \dot{Z}_a \quad (28)$$

and for the second term

$$a D^a u_a \frac{c_s^2}{a(1+\omega)}\Delta^b Z_b = \frac{c_s^2}{1+\omega}\Delta^b Z_b \Theta. \quad (29)$$

Taking the comoving divergence of the first line on the right hand side of Eq. (18) gives

$$-\frac{2}{3}(Z^a Z_a + \Theta Z) - (1+3c_s^2)\frac{1}{2}(\rho\Delta + \rho\Delta^a\Delta_a) - \frac{c_s^2}{1+\omega}\left(a\Delta_a D^a\dot{\Theta} + \dot{\Theta}\Delta\right) - \frac{c_s^2}{1+\omega}D^2\Delta, \quad (30)$$

where we have applied the definition $a D_a\Theta =: Z_a$ as well as $a D^a Z_a =: Z$ and used $a D_a\rho = \rho\Delta_a$. In the second line of Eq. (18), we find for the comoving divergences

$$-\sigma_{ab}Z^{\langle ab\rangle}, \quad -2a^2 D^2\sigma^2 \quad \text{and} \quad +\frac{c_s^4}{(1+\omega)^2}D^2(\Delta^a\Delta_a), \quad (31)$$

where we have used in the first term $a D_b Z_a = 1/3 Z h_{ab} + Z_{\langle ab\rangle} + Z_{[ab]}$. So far we find for the evolution of the volume expansion gradient

$$\dot{Z} = -\frac{2}{3}\frac{c_s^2}{1+\omega}\Theta\Delta^a Z_a - \frac{c_s^2}{1+\omega}\Delta^a\dot{Z}_a$$

$$-\frac{2}{3}(Z^a Z_a + \Theta Z) - \frac{1}{2}(1+3c_s^2)(\rho\Delta + \rho\Delta^a\Delta_a)$$

$$-\frac{c_s^2}{1+\omega}D^2\Delta - 2\sigma_{ab}Z^{\langle ab\rangle} - 2a^2 D^2\sigma^2$$

$$-\frac{c_s^2}{1+\omega}(\Delta_a a D^a\dot{\Theta} + \dot{\Theta}\Delta)$$

$$+\frac{c_s^4}{(1+\omega)^2}D^2(\Delta^a\Delta_a). \quad (32)$$

## 2.2 Second order differential equation for second order perturbations

Taking a further time derivative of Eq. (27) leads to an equation of motion for $\Delta$ which reads

$$\ddot{\Delta} = \omega(\dot{Z}_a\Delta^a + Z^a\dot{\Delta}_a + \dot{\Theta}\Delta + \Theta\dot{\Delta}) - (1+\omega)\dot{Z}$$

$$-2\frac{d}{dt}\left(\sigma_{ab}\Delta^{\langle ab\rangle}\right) - \frac{c_s^2}{1+\omega}\left(\frac{2}{3}\dot{\Theta} + \frac{2}{3}\Theta\frac{d}{dt} + \frac{1}{2}\frac{d^2}{dt^2}\right)\Delta^a\Delta_a. \quad (33)$$

This equation still depends on the volume expansion gradient. In what follows we eliminate this dependence. First, we substitute $\dot{Z}$ from Eq. (32) in Eq. (33) and find

$$
\begin{aligned}
\ddot{\Delta} =& \frac{2}{3}(1+\omega)\Theta Z + \omega\Theta\dot{\Delta} + (\omega + c_s^2)\dot{\Theta}\Delta + \frac{(1+\omega)(1+3c_s^2)}{2}\rho\Delta + c_s^2 D^2\Delta \\
& - 2\frac{d}{dt}\left(\sigma_{ab}\Delta^{\langle ab \rangle}\right) + 2(1+\omega)\sigma_{ab}Z^{\langle ab \rangle} + 2a^2(1+\omega)D^2\sigma^2 \\
& - \frac{c_s^2}{1+\omega}\left[\frac{2}{3}\dot{\Theta} - \frac{(1+\omega)^2(1+3c_s^2)}{c_s^2}\frac{\rho}{2} + c_s^2 D^2 + \frac{2}{3}\Theta\frac{d}{dt} + \frac{1}{2}\frac{d^2}{dt^2}\right]\Delta^a\Delta_a \\
& + (\omega + c_s^2)\dot{Z}^a\Delta_a + (\omega\dot{\Delta}_a + \frac{2}{3}c_s^2\Theta\Delta_a + \frac{2}{3}(1+\omega)Z_a)Z^a + c_s^2(\Delta_a aD^a\dot{\Theta}).
\end{aligned}
\tag{34}
$$

We then replace $\dot{\Theta}$ by the Raychaudhuri equation (A.22)

$$
\dot{\Theta} = -\frac{1}{3}\Theta^2 - \frac{1}{2}(1+3\omega)\rho - 2\sigma^2 - \frac{c_s^2}{a^2(1+\omega)}\Delta + \frac{c_s^4}{a^2(1+\omega)^2}\Delta_a\Delta^a + \Lambda
\tag{35}
$$

and the scalar version of the volume expansion gradient

$$
Z = -\frac{1}{1+\omega}\left(\dot{\Delta} - \omega Z^a\Delta_a - \omega\Theta\Delta + 2\sigma_{ab}\Delta^{\langle ab \rangle} + \frac{c_s^2}{1+\omega}\left(\frac{2}{3}\Theta + \frac{1}{2}\frac{d}{dt}\right)\Delta^a\Delta_a\right),
\tag{36}
$$

which is taken from Eq. (27). The spatial gradient of the Raychaudhuri equation yields

$$
c_s^2\Delta_a aD^a\dot{\Theta} = -c_s^2\frac{2}{3}\Theta\Delta_a Z^a - \frac{c_s^2}{2}(1+3\omega)\rho\Delta_a\Delta^a - \frac{c_s^4}{(1+\omega)a}\Delta_a D^a\Delta.
\tag{37}
$$

Inserting the last three equations into Eq. (34) we find

$$
\begin{aligned}
\ddot{\Delta} &+ \left(\frac{2}{3} - \omega\right)\Theta\dot{\Delta} - \left((\omega - c_s^2)\frac{\Theta^2}{3} + (1 + 2c_s^2 - 3\omega^2)\frac{\rho}{2} + (\omega + c_s^2)\Lambda + c_s^2 D^2\right)\Delta \\
&= -2\left(\frac{2}{3}\Theta + \frac{d}{dt}\right)\left(\sigma_{ab}\Delta^{\langle ab \rangle}\right) + 2(1+\omega)\sigma^{ab}Z_{\langle ab \rangle} + 2a^2(1+\omega)D^2\sigma^2 \\
&\quad - \frac{c_s^2}{1+\omega}\left[\frac{2}{9}\Theta^2 - \frac{1}{c_s^2}\left((1+\omega)^2 + \left(\frac{8}{3} + 4\omega\right)c_s^2\right)\frac{\rho}{2} + c_s^2 D^2 + \frac{2}{3}\Lambda + \Theta\frac{d}{dt} + \frac{1}{2}\frac{d^2}{dt^2}\right]\Delta^a\Delta_a \\
&\quad + (\omega + c_s^2)\dot{Z}^a\Delta_a + \left(\omega\dot{\Delta}_a + \frac{2}{3}\Theta\omega\Delta_a + \frac{2}{3}(1+\omega)Z_a\right)Z^a \\
&\quad - c_s^2\frac{\omega + c_s^2}{(1+\omega)a^2}\Delta^2 - \frac{c_s^4}{(1+\omega)a}\Delta_a D^a\Delta.
\end{aligned}
\tag{38}
$$

Let us emphasize once more that products of three variables for which $S^{(0)} = 0$ are at least of third order and are thus neglected here.

**Perturbative expansion:** So far we have neglected terms that are at least of third order in a perturbative expansion of the dynamical variables. To complete the perturbative analysis, we expand the remaining variables and truncate the series at second order

$$
\Delta \approx \Delta^{(0)} + \Delta^{(1)} + \Delta^{(2)} \equiv \Delta^{(1)} + \Delta^{(2)},
\tag{39}
$$

$$
Z \approx Z^{(0)} + Z^{(1)} + Z^{(2)} \equiv Z^{(1)} + Z^{(2)},
\tag{40}
$$

$$
\sigma \approx \sigma^{(0)} + \sigma^{(1)} + \sigma^{(2)} \equiv \sigma^{(1)} + \sigma^{(2)},
\tag{41}
$$

$$
\Theta \approx \Theta^{(0)} + \Theta^{(1)} + \Theta^{(2)},
\tag{42}
$$

$$
\rho \approx \rho^{(0)} + \rho^{(1)} + \rho^{(2)}.
\tag{43}
$$

Vector and tensor versions of a quantity are expanded in the same manner as their scalar counterparts given above. In Eq. (38) the shear couples either to itself, to $\Delta$ or $Z$ whose zeroth order term is zero. Thus in an expansion to second order only the first order perturbation term of the shear will survive and thus we can immediately put $\sigma \approx \sigma^{(1)}$. Also note that $\rho^{(2)}$ and $\Theta^{(2)}$ will not occur in our perturbed formula because $\rho$ and $\Theta$ always appear in combination with a linearly gauge independent quantity for which the zeroth order term is zero. Expanding the variables in Eq. (38) in this manner we get

$$
\begin{aligned}
\ddot{\Delta}^{(2)} + &\left(\frac{2}{3} - \omega\right)\Theta^{(0)}\dot{\Delta}^{(2)} - \left(\frac{1}{3}(\omega - c_s^2)\Theta^{(0)2} + (1 + 2c_s^2 - 3\omega^2)\frac{\rho^{(0)}}{2} + (\omega + c_s^2)\Lambda + c_s^2 D^2\right)\Delta^{(2)} \\
= &-\left(\frac{2}{3} - \omega\right)\Theta^{(1)}\dot{\Delta}^{(1)} + \left(\frac{2}{3}(\omega - c_s^2)\Theta^{(0)}\Theta^{(1)} + (1 + 2c_s^2 - 3\omega^2)\frac{\rho^{(1)}}{2}\right)\Delta^{(1)} \\
&- 2\left(\frac{2}{3}\Theta^{(0)} + \frac{d}{dt}\right)\left(\sigma_{ab}^{(1)}\Delta^{(1)\langle ab\rangle}\right) + 2(1+\omega)\sigma^{(1)ab}Z_{\langle ab\rangle}^{(1)} + 2a^2(1+\omega)D^2\sigma^{(1)2} \\
&- \frac{c_s^2}{1+\omega}\left[\frac{2}{9}\Theta^{(0)2} - \frac{1}{c_s^2}\left((1+\omega)^2 + \left(\frac{8}{3} + 4\omega\right)c_s^2\right)\frac{\rho^{(0)}}{2}\right. \\
&\qquad\qquad\left. + c_s^2 D^2 + \frac{2}{3}\Lambda + \Theta^{(0)}\frac{d}{dt} + \frac{1}{2}\frac{d^2}{dt^2}\right]\Delta^{(1)a}\Delta_a^{(1)} \\
&+ (\omega + c_s^2)\dot{Z}^{(1)a}\Delta_a^{(1)} + \left(\omega\dot{\Delta}_a^{(1)} + \frac{2}{3}\Theta^{(0)}\omega\Delta_a^{(1)} + \frac{2}{3}(1+\omega)Z_a^{(1)}\right)Z^{(1)a} \\
&- c_s^2\frac{\omega + c_s^2}{(1+\omega)a^2}\Delta^{(1)2} - \frac{c_s^4}{(1+\omega)a}\Delta_a^{(1)}D^a\Delta^{(1)} .
\end{aligned}
$$
(44)

We have ordered the terms in this equation such that second order quantities appear on the left hand side of the equation and combinations of first order variables appear on the right hand side. At this point it is also useful to replace the contractions with $Z_a^{(1)}$ and $\dot{Z}_a^{(1)}$. To do so, we use Eq. (9) which yields

$$
Z_a = \frac{\omega\Theta\Delta_a - \dot{\Delta}_{\langle a\rangle} - \sigma_{ab}\Delta^b}{1+\omega} ,
$$
(45)

$$
\Rightarrow Z_a^{(1)} = \frac{\omega\Theta^{(0)}\Delta_a^{(1)} - \dot{\Delta}_a^{(1)}}{1+\omega} + \mathcal{O}(\epsilon^2) .
$$
(46)

Similarly, from Eqs. (18) and (46) we get for the time derivative of $Z_a^{(1)}$

$$
\begin{aligned}
\dot{Z}_a^{(1)} = &\frac{1}{3}\frac{\Theta^{(0)2}}{1+\omega}\left(c_s^2 - 2\omega\right)\Delta_a^{(1)} + \frac{2}{3}\frac{1}{1+\omega}\Theta^{(0)}\dot{\Delta}_a^{(1)} - \frac{c_s^2}{a(1+\omega)}D^a\Delta^{(1)} \\
&+ \left(\frac{c_s^2}{1+\omega}(1+3\omega) - (1+3c_s^2)\right)\frac{\rho^{(0)}}{2}\Delta_a^{(1)} .
\end{aligned}
$$
(47)

The respective terms in Eq. (44) become

$$(\omega + c_s^2)\dot{Z}^{(1)a}\Delta_a^{(1)} = \frac{1}{3}\frac{\omega + c_s^2}{1+\omega}(c_s^2 - 2\omega)\Theta^{(0)2}\Delta_a^{(1)}\Delta^{(1)a}$$

$$+ \frac{1}{3}\frac{\omega + c_s^2}{1+\omega}\Theta^{(0)}\frac{d}{dt}(\Delta^{(1)a}\Delta_a^{(1)}) - \frac{c_s^2(c_s^2 + \omega)}{a(1+\omega)}\Delta_a^{(1)}D^a\Delta^{(1)}$$

$$+ (\omega + c_s^2)\left(\frac{c_s^2}{1+\omega}(1+3\omega) - (1+3c_s^2)\right)\frac{\rho^{(0)}}{2}\Delta_a^{(1)}\Delta^{(1)a}, \qquad (48)$$

and

$$\left(\omega\dot{\Delta}_a^{(1)} + \frac{2}{3}\Theta^{(0)}\omega\Delta_a^{(1)} + \frac{2}{3}(1+\omega)Z_a^{(1)}\right)Z^{(1)a} =$$

$$\frac{\omega - 2}{2}\frac{\omega}{1+\omega}\Theta^{(0)}\frac{d}{dt}(\Delta^{(1)a}\Delta_a^{(1)}) + \frac{\frac{2}{3} - \omega}{1+\omega}\dot{\Delta}^{(1)a}\dot{\Delta}_a^{(1)} + \frac{4}{3}\frac{\omega^2}{1+\omega}\Theta^{(0)2}\Delta^{(1)a}\Delta_a^{(1)}. \quad (49)$$

With these we can eliminate $\dot{Z}_a$ and $Z_a$ completely and our perturbed second order differential equation for second order density perturbations in a perfect fluid with shear reads

$$\ddot{\Delta}^{(2)} + \left(\frac{2}{3} - \omega\right)\Theta^{(0)}\dot{\Delta}^{(2)} - \left(\frac{1}{3}(\omega - c_s^2)\Theta^{(0)2} + (1+2c_s^2 - 3\omega^2)\frac{\rho^{(0)}}{2} + (\omega + c_s^2)\Lambda + c_s^2 D^2\right)\Delta^{(2)}$$

$$= -\left(\frac{2}{3} - \omega\right)\Theta^{(1)}\dot{\Delta}^{(1)} + \left(\frac{2}{3}(\omega - c_s^2)\Theta^{(0)}\Theta^{(1)} + (1+2c_s^2 - 3\omega^2)\frac{\rho^{(1)}}{2}\right)\Delta^{(1)}$$

$$- 2\left(\frac{2}{3}\Theta^{(0)} + \frac{d}{dt}\right)\left(\sigma_{ab}^{(1)}\Delta^{(1)\langle ab\rangle}\right) + 2(1+\omega)\sigma_{ab}^{(1)}Z^{(1)\langle ab\rangle} + 2a^2(1+\omega)D^2\sigma^{(1)2}$$

$$- \frac{c_s^2}{1+\omega}\left[\left(\frac{2}{3} - \frac{2\omega^2 + c_s^4 - \omega c_s^2}{c_s^2}\right)\frac{\Theta^{(0)2}}{3} + \frac{1}{c_s^2}\left(-1 - \omega - \omega c_s^2 - \frac{5}{3}c_s^2 + 2c_s^4\right)\frac{\rho^{(0)}}{2}\right.$$

$$\left.+ c_s^2 D^2 + \frac{2}{3}\Lambda + \left(\frac{2}{3} + \frac{2\omega}{3c_s^2} - \frac{\omega^2}{2c_s^2}\right)\Theta^{(0)}\frac{d}{dt} + \frac{1}{2}\frac{d^2}{dt^2}\right]\Delta^{(1)a}\Delta_a^{(1)}$$

$$+ \frac{\frac{2}{3} - \omega}{1+\omega}\dot{\Delta}_a^{(1)}\dot{\Delta}^{(1)a} - \frac{c_s^2(c_s^2 + \omega)}{(1+\omega)a^2}\Delta^{(1)2} - \frac{c_s^2(2c_s^2 + \omega)}{(1+\omega)a}\Delta_a^{(1)}D^a\Delta^{(1)}. \qquad (50)$$

We fix the background cosmology by introducing a further assumption.

**Assumption 4:** As background we choose an FLRW universe: $\Theta^{(0)}(t) = 3H(t)$ and $\rho^{(0)} = \rho(t)$ is given by the Friedmann and continuity equations

$$\dot{H} = -H^2 - \frac{1}{6}(\rho + 3p) + \frac{1}{3}\Lambda, \quad \dot{\rho} = -3H(\rho + p), \text{ and}$$

$$H^2 = \frac{1}{3}\rho - \frac{K}{a^2} + \frac{1}{3}\Lambda.$$

On this background our equation eventually yields

$$\ddot{\Delta}^{(2)} + 2H\left(1 - \frac{3}{2}\omega\right)\dot{\Delta}^{(2)}$$

$$- \left[\frac{3}{2}\left(1 + 2\omega - 3\omega^2\right)H^2 - \frac{1}{2}\left(1 - 2\omega - 3\omega^2\right)\Lambda + \left(1 + 2c_s^2 - 3\omega^2\right)\frac{3K}{2a^2} + c_s^2 D^2\right]\Delta^{(2)}$$

$$= -\left(\frac{2}{3} - \omega\right)\Theta^{(1)}\dot{\Delta}^{(1)} + \left(\frac{1}{3}(\omega - c_s^2)6H\Theta^{(1)} + (1+2c_s^2 - 3\omega^2)\frac{\rho^{(1)}}{2}\right)\Delta^{(1)}$$

$$- 2\left(2H + \frac{d}{dt}\right)\sigma_{ab}^{(1)}\Delta^{(1)\langle ab\rangle} + 2(1+\omega)\sigma_{ab}^{(1)}Z^{(1)\langle ab\rangle} + 2(1+\omega)a^2 D^2\sigma^{(1)2}$$

$$-\frac{c_s^2}{1+\omega}\left[\left(-1-\omega+\frac{7}{3}\omega c_s^2-\frac{5}{3}c_s^2-4\omega^2\right)\frac{\rho^{(0)}}{2c_s^2}+c_s^2\mathrm{D}^2+\left(\omega-2\frac{\omega^2}{c_s^2}-c_s^2+\frac{2}{3}\right)\Lambda\right.$$
$$\left.+\left(\frac{2}{3}+\frac{2\omega}{3c_s^2}-\frac{\omega^2}{2c_s^2}\right)3H\frac{\mathrm{d}}{\mathrm{d}t}+\frac{1}{2}\frac{\mathrm{d}^2}{\mathrm{d}t^2}\right]\Delta^{(1)a}\Delta_a^{(1)}$$
$$+\frac{\frac{2}{3}-\omega}{1+\omega}\dot{\Delta}_a^{(1)}\dot{\Delta}^{(1)a}-\frac{c_s^2(c_s^2+\omega)}{(1+\omega)a^2}\Delta^{(1)2}-\frac{c_s^2(2c_s^2+\omega)}{(1+\omega)a}\Delta_a^{(1)}\mathrm{D}^a\Delta^{(1)}. \tag{51}$$

In this equation the second order density perturbations on the left hand side are sourced by couplings of first order perturbations on the right hand side. In particular, we find source terms in which the shear tensor couples to first order density perturbations but also to itself. The next step is to study our equation in the two important regimes where either radiation or matter dominates. For that we reintroduce $\kappa := 8\pi G$.

**Matter dominated universe:**  For a matter dominated universe we have $\omega = c_s^2 = 0$. The linearized evolution equation for the shear tensor and the Gauss-Codazzi equation [29] determine the projected Ricci tensor by the shear $R_{\langle ab\rangle} = -3H\sigma_{ab}-\dot{\sigma}_{ab}$. For super-horizon shear modes $\mathrm{D}^2\sigma^2 = 0$ in a flat universe $K = 0$, this identity together with the relations $\dot{\Delta}_{\langle ab\rangle} = -Z_{\langle ab\rangle}$ and $\dot{\Delta}_a = -Z_a$ (details see ref. [29]) reduces Eq. (51) to

$$\ddot{\Delta}^{(2)}+2H\dot{\Delta}^{(2)}-\frac{1}{2}(3H^2-\Lambda)\Delta^{(2)}=\frac{3H^2}{2}\kappa\Delta_a^{(1)}\Delta^{(1)a}+\frac{2}{3}Z_a^{(1)}Z^{(1)a}+2H\sigma_{ab}^{(1)}\Delta^{(1)\langle ab\rangle}$$
$$+4\sigma_{ab}^{(1)}Z^{(1)\langle ab\rangle}+2R_{\langle ab\rangle}^{(1)}\Delta^{(1)\langle ab\rangle}$$
$$-\frac{2}{3}\Theta^{(1)}\dot{\Delta}^{(1)}+\frac{1}{2}\rho^{(1)}\Delta^{(1)}. \tag{52}$$

which partially reproduces the result in [29] for $\Lambda = 0$. We find two additional terms $-2H\Theta^{(1)}\dot{\Delta}^{(1)}+\frac{1}{2}\rho^{(1)}\Delta^{(1)}$ which were missed by the reduction procedure used in [29].

**Radiation dominated universe:**  Important for this work is the evolution of the perturbations during radiation domination where we have $\omega = c_s^2 = 1/3$. For our purpose it is also convenient to eliminate $Z_{\langle ab\rangle}$ such that we get an equation only depending on $\sigma_{ab}$ and $\Delta$ in its various forms. We also take a flat universe with $K = 0$ and $\Lambda = 0$. Again by using Eq. (17) the volume expansion gradients yield to linear order $(1+\omega)Z_a^{(1)} = \omega\Theta\Delta_a^{(1)}-\dot{\Delta}^{(1)}$. Taking the comoving derivative $a\mathrm{D}^b$ of the previous expression, using the decomposing rule $a\mathrm{D}_b Z_a = 1/3 h_{ab}Z+Z_{\langle ab\rangle}+Z_{[ab]}$ (analogously for $\Delta_a$) and the linear rule $a\mathrm{D}_b\dot{\Delta}_a^{(1)} = a\frac{\mathrm{d}}{\mathrm{d}t}(\mathrm{D}_b\Delta_a^{(1)})$ we can estimate

$$Z_{\langle ab\rangle}^{(1)} = \frac{\omega\Theta^{(0)}\Delta^{(1)}h_{ab}+\omega\Theta^{(0)}\Delta_{\langle ab\rangle}^{(1)}-\dot{\Delta}_{\langle ab\rangle}^{(1)}}{1+\omega}+\mathcal{O}(\epsilon^2). \tag{53}$$

Finally our equation yields in a flat universe without cosmological constant

$$\ddot{\Delta}^{(2)}+H\dot{\Delta}^{(2)}-2H^2\Delta^{(2)}-\frac{1}{3}\mathrm{D}^2\Delta^{(2)}$$
$$=-2\left(H\sigma_{ab}^{(1)}\Delta^{(1)\langle ab\rangle}+\dot{\sigma}_{ab}^{(1)}\Delta^{(1)\langle ab\rangle}+2\sigma_{ab}^{(1)}\dot{\Delta}^{(1)\langle ab\rangle}\right)+\frac{8}{3}a^2\mathrm{D}^2\sigma^{(1)2} \quad\}\;\text{GW sources}$$
$$-\frac{1}{3}\Theta^{(1)}\dot{\Delta}^{(1)}+\frac{2}{3}\rho^{(1)}\Delta^{(1)}$$
$$+\left(2H^2-\frac{1}{12}\mathrm{D}^2\right)\Delta_a^{(1)}\Delta^{(1)a}-\frac{7}{4}H\dot{\Delta}_a^{(1)}\Delta^{(1)a}-\frac{1}{4}\ddot{\Delta}_a^{(1)}\Delta^{(1)a} \quad\}\;\text{Pure density sources}$$
$$-\frac{1}{6}\frac{1}{a^2}\Delta^{(1)2}-\frac{1}{4}\frac{1}{a}\Delta_a^{(1)}\mathrm{D}^a\Delta^{(1)}. \tag{54}$$

# 3 First order phase transitions and GWs

First order phase transitions occur when a configuration does not minimize the energy anymore. In this process the minimum $\langle\phi\rangle$ (order parameter) of the temperature dependent potential $V_T(\phi)$ evolves from a symmetric (unordered) phase $\langle\phi\rangle = 0$ to an asymmetric (ordered) phase $\langle\phi\rangle \neq 0$. If the former minimum and the new minimum in the potential are separated by a potential barrier then the order parameter does not smoothly roll into the new minimum but rather jumps or tunnels (see Fig. 1). The barrier of a FPT prevents the system from continuously relaxing into a new state with lower energy which results in latent heat being stored and later released in a shorter time interval. This delayed energy releasable makes FPTs particularly interesting in comparison to other phase transitions. The temperature at which the two minima are degenerate is called the critical temperature $T_c$ while the temperature at which the probability per volume element of reaching the new minimum is unity is called nucleation temperature $T_{\mathrm{nuc}}$.

In the early universe such phase transitions could have happened and are realized in many extensions of the SM [5–10, 12, 14–17] where a Lagrangian $\mathcal{L}(\{\phi_i\})$ with extra fields $\phi_i$, symmetries and couplings is introduced. Here the order parameter is the vacuum expectation value (VEV) of the field which acquires a non zero mass in the low temperature phase. This results in a "spontaneous breaking" of the involved symmetries, i.e. a non-linear realization of the symmetry in the vacuum at zero temperature. In a FPT the field does not accept the new VEV everywhere in space at the same time. Instead bubbles with the new VEV inside nucleate with some initial sizes and begin to spatially expand into regions formerly occupied by the high temperature, symmetric VEV. Hereby the bubbles release the stored energy, the latent heat fraction

$$\alpha = \frac{\rho_{\mathrm{vac}}}{\rho_{\mathrm{rad}}(T_{\mathrm{nuc}})}, \tag{55}$$

where $\rho_{\mathrm{vac}} \sim |V_{T_{\mathrm{nuc}}}(\langle\phi\rangle)|$, and release it in the form of motion but also while their surfaces, called walls, eventually collide. The time needed until the field $\phi$ has acquired the new VEV everywhere in the universe is denoted by $1/\beta$ (see Fig. 1 second plot) and is the inverse of the nucleation rate per Hubble volume

$$\Gamma = \Gamma_{\mathrm{nuc}} \exp(\beta(t - t_{\mathrm{nuc}})). \tag{56}$$

The rate $\beta$, in turn, is deduced from the $O(3)$-symmetric, effective action [31]

$$S_3 = 4\pi \int_0^\infty r^2 \mathrm{d}r \left( \frac{1}{2}\left(\frac{\mathrm{d}\phi}{\mathrm{d}r}\right)^2 + V_T(\phi) \right), \quad \text{via} \quad \frac{\beta}{H_{\mathrm{nuc}}} = T_{\mathrm{nuc}} \frac{\mathrm{d}S_3}{\mathrm{d}T}\bigg|_{T_{\mathrm{nuc}}}, \tag{57}$$

whose minimization determines the tunneling trajectory from the former, symmetric vacuum to the low temperature, forming vacuum. In this way, via the form of the potential $V_T(\phi) \subset \mathcal{L}(\phi)$, the phase transition parameters $\alpha$ and $\beta$ are directly connected to the details of the particle physics model. Whether or not a certain model posseses a potential barrier and a FPT depends therefore on both the model details and the choice of coupling parameters.

The formation of bubbles has an important implication. By their expansion and collision, they induce three different forms of anisotropic stress into the fluid, which in turn generate GWs.

1. Bubble collision: The collision of the forming and expanding bubbles leads to an anisotropic stress that sources GWs [32–34],

2. Turbulence: The highly ionized plasma can develop magnetic and hydrodynamical turbulence from the percolation of the fluid induced by the bubble collisions [35, 36],

3. Sound waves: The bulk fluid motion produces pressure waves in the fluid that source GWs [37–39],

such that the total abundance of GWs from a FPT reads

$$\Omega_{\text{GW}}(k) = \Omega_{\text{BC}}(k) + \Omega_{\text{SW}}(k) + \Omega_{\text{MHD-turb}}(k). \tag{58}$$

In summary the important parameters are:

- The *nucleation temperature* $T_{\text{nuc}}$ ($\sim H_{\text{nuc}}$) which sets the scale of the released energy density in GWs $\rho_{\text{GW}}$.

- The *strength* of the phase transition is described by the latent heat fraction $\alpha$ and its *duration* by the inverse nucleation rate $\beta^{-1}$.

The strength of the GW signal also depends on the *bubble wall velocity* $v_w$. Its calculation is more involved (see eg. [16, 40–42]). Depending on the bubble dynamics not all of the released energy produces GWs. The fraction of the released energy actually transmitted to the kinetic energy of the fluid is provided by the *efficiency factor* $\kappa_{\text{eff}}$. If the phase transition does not reheat the universe too much one can approximate the temperature at which the GWs are released as $T_* \approx T_{\text{nuc}}$. On the other hand in models with large $\alpha > 1$ there will be a supercooling phase which separates the nucleation temperature from the percolation temperature $T_{\text{nuc}} \ll T_*$ [16]. The release of GWs then falls together with reheating of the universe $T_* = T_{\text{reh}}$ which turns the universe back into radiation domination. If the reheating is fast enough we do not need to distinguish between $H(T_{\text{nuc}})$ and $H_*$ [17]. The two cases constitute very different bubble dynamics. In the former the contribution from bubble walls to the GW signal is only relevant for so called run-away dynamics in which the bubble walls strongly accelerate until they reach the speed of light. However, they expand into a still radiation dominated universe such that parts of the available energy are absorbed by the plasma and thus the efficiency factor $\kappa_{\text{eff}}$ is smaller than unity. In the latter case, where the bubbles propagate with the speed of light $v_w = c$ in a vacuum energy dominated universe, the efficiency factor equals unity and bubble walls are the only contribution to GW production. The efficiency factor thus reads [16]

$$\kappa_{\text{eff}} = 1 \quad \text{for} \quad \alpha > 1. \tag{59}$$

Note, that for the purpose of this paper only the time and scale of GW production is important and therefore we will always refer to $t_*$ and not bother about $t_{\text{nuc}}$, also in the case of strong supercooling because in any case $H(t_*) \approx H(t_{\text{nuc}})$.
The sufficient set of parameters describing the phase transition is then $(H_*, \beta, \alpha, \kappa_{\text{eff}}(\alpha))$.

## 3.1 Analytic description of bubble collision

We study the GW energy density originating from bubble collisions as source of second order density perturbations following the literature, e.g. [32, 34, 40, 43, 44]. Typical central assumptions in these derivations are:

1. **Thin wall:** the bubble walls are infinitesimal thin and all energy is stored on them,

2. **Envelope approximation:** Already collided walls do not source GWs. Only the remaining envelope of collided bubbles carries energy and momentum.

3. The phase transition performs in **less than a Hubble time**.

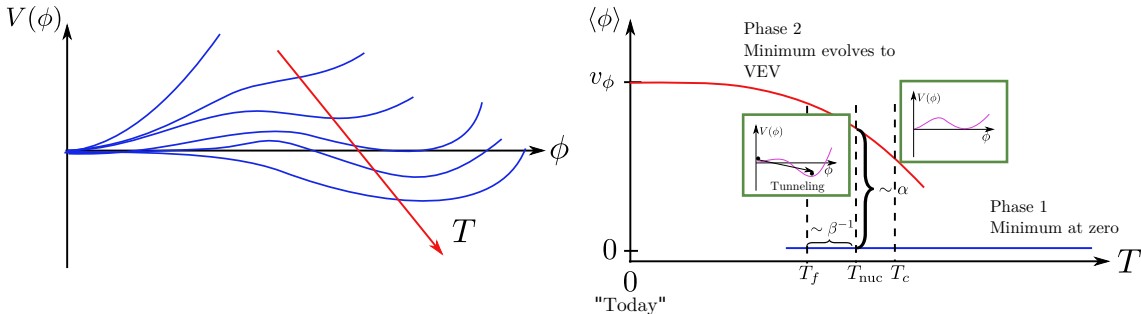

Figure 1: **Left:** Qualitative temperature dependence of a typical potential in particle physics developing a barrier and new global minimum as temperature drops. **Right:** Schematic illustration of the temperature dependence of the VEV (order parameter) in a FPT. Shown are also the two important temperatures $T_c$ and $T_{\text{nuc}}$ at which the minima are degenerated and at which the nucleation probability reaches one bubble per Hubble volume, respectively. The temperature at which the transition is completed is denoted by $T_f$.

GWs originate from linear tensor perturbations (for consistency with the literature in this subsection Latin indices are spatial and run from 1 to 3.)

$$ds^2 = -dt^2 + a^2(t)(\delta_{ij} + 2h_{ij})dx^i dx^j, \tag{60}$$

by a tracefree and transverse tensor $h_{ij}(\mathbf{x}, t)$. The GWs propagate according to the wave equation [45,46] and are sourced by the transverse and tracefree component of the anistropic stress tensor $\Pi_{ij}^{\perp}(\mathbf{x}, t)$. In Fourier space ($k$ denotes comoving wave number) the equation of motion reads

$$\ddot{h}_{ij}(\mathbf{k}, t) + \frac{k^2}{a^2} h_{ij}(\mathbf{k}, t) = 16\pi G\, \Pi_{ij}^{\perp}(\mathbf{k}, t). \tag{61}$$

Solving this equation for a given anisotropic stress tensor allows to derive the energy density of GWs

$$\rho_{\text{GW}}(t) := \frac{\langle \dot{h}_{ij} \dot{h}_{ij} \rangle}{8\pi G} = \int_0^{\infty} \frac{k^3}{2\pi} |\dot{h}(k, t)|^2 d\ln k. \tag{62}$$

In the case of FPTs, the collision of bubbles produces an anistropic stress tensor $\Pi_{ij}^{\perp}(\mathbf{k}, t)$ which drives GWs through Eq. (61). This leads to an energy density per logarithmic frequency, which is given by (since the FPT is short we can put $a(t) \approx a_*$)

$$\Omega_{\text{GW}}^{\log}(k/a_*, t) := \frac{1}{\rho_{\text{tot}}} \frac{d\rho_{\text{GW}}}{d\ln k} = \kappa_{\text{eff}}^2 \left(\frac{H_*}{\beta}\right)^2 \left(\frac{\alpha}{1+\alpha}\right)^2 \Delta(k/a_*, \beta, t, v_w), \tag{63}$$

where $\kappa_{\text{eff}}$ is the efficiency factor. The challenge for analytical [32,33,43,44] and numerical studies [47–49] is to find an expression for the dimensionless power spectrum $\Delta(k/\beta, t, v_w)$. The essential ingredients are the homogeneous solution of the wave equation Eq. (61) and the power spectrum of the anisotropic stress tensor evaluated at different times. Here we simply refer to the literature and stick to an approximate formula from Caprini et al. [40] which incorporates the most important features[2] . Following this reference the dimensionless power spectrum is well described by

---

[2]In fact the approximation seems very close to more refined analyses like [44]. The latter reference explicitly mentions that the underling assumptions listed above are especially well fulfilled for bubbles expanding into vacuum. This is in particular important for this work since significant impact will only be generated in this regime.

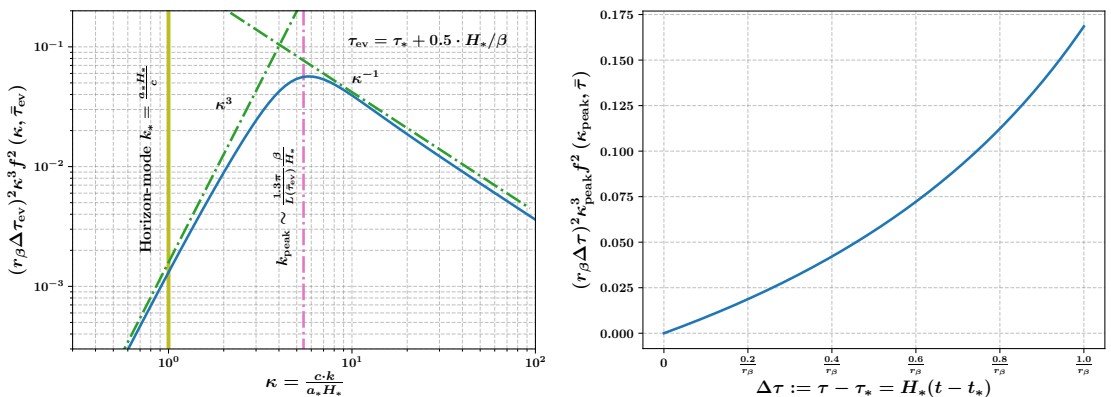

Figure 2: **Left:** The dimensionless power spectrum (with approximated time integral) of GWs sourced by a FPT evaluated at halftime $\tau = \tau_* + 0.5 \cdot H_*/\beta$. **Right:** The time evolution of the peak of the dimensionless power spectrum. The inverse duration in terms of Hubble time is denoted by $r_\beta = \frac{\beta}{H_*}$ and $\bar{\tau}_{ev} := \frac{\tau_{ev} + \tau_*}{2}$ is the time mean while $\Delta\tau_{ev} := \tau_{ev} - \tau_*$.

$$\Delta(k, \beta, t, v_w) = \beta^2 k^3 \left| \int_{t_*}^t f(k, \tilde{t}) e^{i\tilde{t}k} \mathrm{d}\tilde{t} \right|^2 \approx k^3 \beta^2 (t - t_*)^2 f^2(k, (t + t_*)/2), \qquad (64)$$

with the broken rational function

$$f(k, t)^2 = L(t)^2 \left( \frac{v_w \epsilon}{\beta} \right) \left( \frac{1 + (\frac{kL}{3})^2}{1 + (\frac{kL}{2})^2 + (\frac{kL}{3})^6} \right) \qquad (65)$$

and the characteristic length- and time-functions

$$L(t) = \frac{v_w}{\beta} g(t), \qquad (66)$$

$$g(t) = 4\beta^2 (t - t_*) \left( \frac{1}{\beta} - (t - t_*) \right) \left[ \Theta_{\mathrm{Hv}}(t - t_*) \cdot \Theta_{\mathrm{Hv}} \left( \frac{1}{\beta} - t_* \right) \right], \qquad (67)$$

where $\epsilon$ is a small parameter (taken to 0.01 in the following), $v_w$ is the bubble expansion speed, $t_*$ the starting time of the release of GWs and $\Theta_{\mathrm{Hv}}(t)$ is the Heaviside step-function. In Fig. 2 we show the dimensionless power spectrum achieved from these functions.

In terms of the rescaled time $\tau := H_* \cdot t$, rescaled Fourier mode $\kappa := ck/(a_* H_*)$ (note that we put $k \to k/a_*$) and the relative phase transition duration $H_*/\beta$ we get (also reintroducing the speed of light $c$)

$$k/a_* \cdot L(t) = \frac{k}{a_* H_*} \cdot H_* \cdot \frac{c}{\beta} \left( \frac{v_w}{c} \right) g(t) = \kappa \left( \frac{v_w}{c} \right) \left( \frac{H_*}{\beta} \right) \tilde{g}(\tau) = \kappa L(\tau), \qquad (68)$$

where the time-dependent function $\tilde{g}(\tau)$ and the broken rational function $f(\kappa, \tau)$ becomes

$$\tilde{g}(\tau) = 4 \left( \frac{\beta}{H_*} \right)^2 (\tau - \tau_*) \left( \left( \frac{H_*}{\beta} \right) - (\tau - \tau_*) \right) \Theta_{\mathrm{Hv}} \left( \tau_*, \tau_* + \frac{H_*}{\beta} \right), \qquad (69)$$

$$k^3 f(k, t) = \kappa^3 f^2(\kappa, \tau) = (\kappa L(\tau))^2 \left( \kappa \frac{v_w}{c} \frac{H_*}{\beta} \epsilon \right) \left( \frac{1 + \left( \frac{\kappa L(\tau)}{3} \right)^2}{1 + \left( \frac{\kappa L(\tau)}{2} \right)^2 + \left( \frac{\kappa L(\tau)}{3} \right)^6} \right). \qquad (70)$$

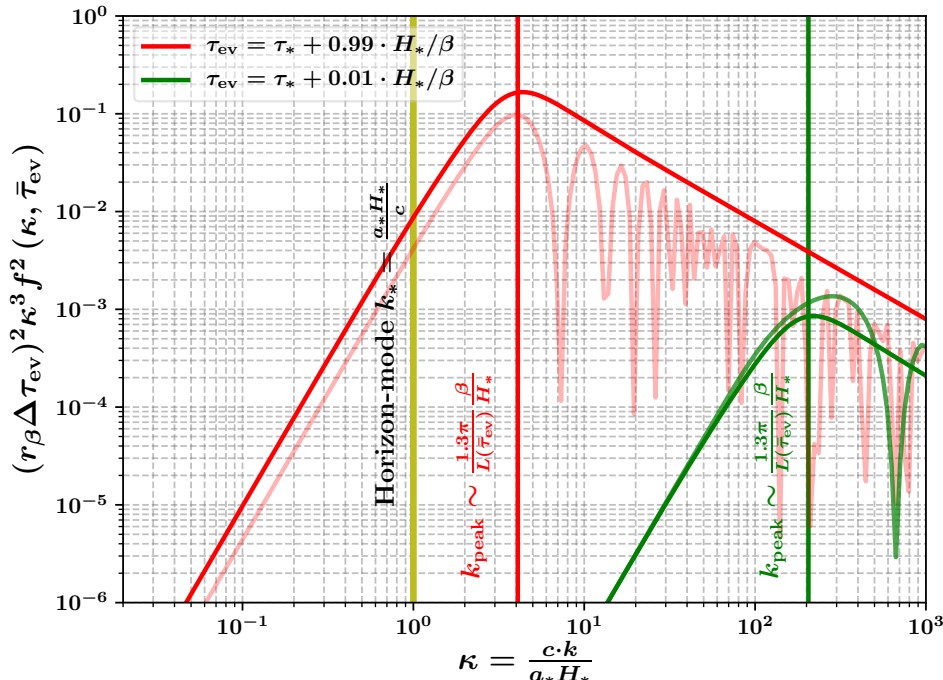

Figure 3: The dimensionless power spectrum of GWs sourced by a FPT (light red and light green) and the approximation of the time integral in Eq. (64) (bold lines) close to the start and the end of the FPT. The yellow curve shows the horizon mode. The notation is the same as in Fig. 2.

Therefore, the rescaled function for the GW abundance is

$$\Omega_{\text{GW}}(\kappa, \tau) = \kappa_{\text{eff}}^2 (\tau - \tau_*)^2 \left( \frac{\alpha}{1 + \alpha} \right)^2 \int_{\kappa_*}^{\kappa} d\tilde{\kappa}\, \tilde{\kappa}^2 f^2(\tilde{\kappa}, (\tau + \tau_*)/2), \tag{71}$$

with $k_* := a_* H_* / c$ the scale of the horizon at $t_*$.

The left plot in Fig. 2 shows the spectrum as a function of time for a fixed mode while the right plot shows the time evolution of the peak $\Delta(\kappa_{\text{peak}}(\tau), \tau)$. In Fig. 3 we show the approximation Eq. (64) for different times. The main three features of the power spectrum of GWs from FPT are

- It peaks around $\sim k_{\text{peak}} = \frac{1.3 \pi \beta}{c \cdot L(\tau_{(\text{ev}} + \tau_*)/2)}$ which is $\frac{2 \pi \beta}{c}$ at the end of the transition,

- For small wave numbers the spectrum grows as $k^3$,

- For large wave numbers the spectrum decreases as $k^{-1}$.

## 4 Results

In this section we present the results obtained from Eq. (54) for the following scenario (see also sketch 4):

During radiation domination a FPT is triggered at time $t_{\text{nuc}}$ and emits GWs by bubble collision at time $t_*$ on sub-horizon scales $k \gg a_* H_*$. The GWs manifest themselves as shear perturbations. The transition completes within less than a Hubble time $t_* + 1/\beta$, where $\beta > H(t_*)$.

The shear distortions induce second order density perturbations via Eq. (54). After sourcing the induced density perturbations remain imprinted in the spectrum. Hence we need to identify the source terms, calculate the density perturbations they induce and transfer them to the matter power spectrum in order to estimate their impact on structure formation.

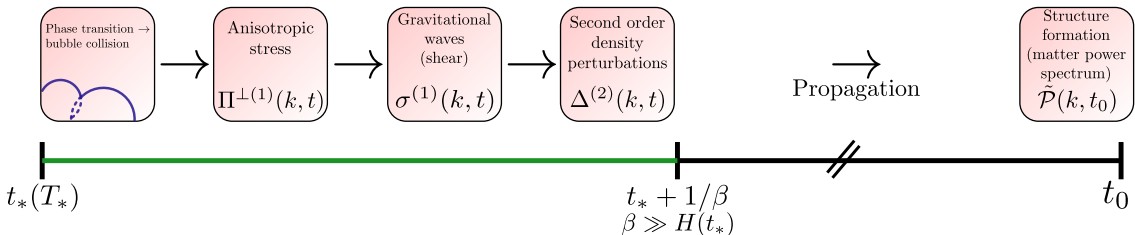

Figure 4: Timeline and schematic flow chart of the model described in the text. The green line depicts the time of the phase transition and the radiation of GWs. The flow chart on top of the line shows the processes happening during that phase on sub-horizon scales. We assume that these processes happen during the whole period of the transition. After the transition is completed, the induced perturbations affect the matter power spectrum today. The two angled lines indicate that the time between the end of the phase transition and today is much longer than the duration of the transition.

To do so, we have to use the solution of Eq. (54) to derive the transfer function $T(k)$. In order to induce any changes at all, the FPT must be strong enough such that the terms not involving the shear tensor are subdominant. Moreover, as we will see in Eq. (75) the shear is related to the GW abundance at the transition time via $|\sigma^{(1)}| \sim H_* \sqrt{3\Omega_{\mathrm{GW}}(\kappa, \tau)}$ and the first order perturbations can be estimated as $|\Delta^{(1)}| \sim 10^{-4}$, see Eq. (103). This implies

$$|\Delta^{(1)}|^2 \sim 10^{-8} \qquad \text{and} \qquad \sigma_{ab}\Delta^{\langle ab \rangle} \sim |\sigma^{(1)}| \times 10^{-4}. \tag{72}$$

Therefore we find the pure shear term to be the most interesting and powerful source term if $|\sigma^{(1)}| > |\Delta^{(1)}| \sim 10^{-4}$. The latter condition is fulfilled for a relatively wide range of phase transition parameters. For $\alpha \to \infty$ the duration can be as small as $\beta/H_* \approx 100$ until the high-$\kappa$ plateau of $\rho_{\mathrm{GW}}$ reaches a magnitude of $10^{-4}$.

Thus, in the following we focus on the self coupling of the shear, namely

$$\ddot{\Delta}^{(2)} + H\dot{\Delta}^{(2)} - 2H^2 \Delta^{(2)} - \frac{1}{3}D^2\Delta^{(2)} = \frac{8}{3}a^2 D^2 \sigma^{(1)2}. \tag{73}$$

Following references [28, 50, 51] the shear tensor is related to the linear, tracefree and transverse metric perturbation by $\sigma_{ab} = a(h_{\alpha\beta})'$ and $\sigma^{ab} = a^{-3}(h_{\alpha\beta})'$ (see also Eq. (A.48)). Recall that in this work $a, b = 0, 1, 2, 3$ and $\alpha, \beta = 1, 2, 3$ and the prime denotes derivative with respect to conformal time. Using the definition of the energy density of GWs

$$\rho_{\mathrm{GW}}(\mathbf{x}, t) = \frac{(h_{\alpha\beta})'(\mathbf{x}, t)(h^{\alpha\beta})'(\mathbf{x}, t)}{2a^2 \, 8\pi G} = \frac{\dot{h}_{\alpha\beta}(\mathbf{x}, t)\dot{h}^{\alpha\beta}(\mathbf{x}, t)}{16\pi G}, \tag{74}$$

we observe that the squared shear $\sigma^2(\mathbf{x}, t) = 1/2\,\sigma_{ab}\sigma^{ab}$ is nothing but

$$\rho_{\mathrm{GW}}(\mathbf{x}, t) = \frac{1}{8\pi G}\sigma^2(\mathbf{x}, t). \tag{75}$$

We can also transform the divergence of the fractional energy density gradient into a more familiar variable. To do so, we note, that in a vorticity free and spatially flat space ($K = 0$) the

projected derivatives become spatial Laplacians

$$D^a D_a = \frac{\nabla^2}{a^2} \tag{76}$$

and thus the divergence of the fractional density gradient can be written as the Laplacian of some function $\tilde{\delta}$ which depends on the relative energy density perturbation $\delta := \delta\rho/\bar{\rho}$

$$\Delta(\mathbf{x}, t) = \nabla^2 \tilde{\delta}(\mathbf{x}, t). \tag{77}$$

If $\Delta(\mathbf{x}, t) \equiv \Delta^{(1)}(\mathbf{x}, t)$ is a linear perturbation then $\tilde{\delta}$ is equivalent to Bardeen's variable for the relative energy density perturbation $\tilde{\delta}^{(1)} := \delta + 3H(1 + \omega)\xi^0$ with the time component $\xi^0$ of an arbitrary gauge transformation $x^\mu \to x^\mu + \xi^\mu$ [52] (see also subsection A.6). Therefore, using Eq. (75) and Eq. (77) the $1 + 3$ covariant variables can be expressed in a more standard manner.

Similarly, in a spatially flat spacetime the harmonic decomposition of the variables reduces to standard Fourier modes (see [52, 53])

$$f(\mathbf{x}, t) = \int_{\mathbf{k}} f_{\mathbf{k}} e^{i\mathbf{k}\cdot\mathbf{x}}, \tag{78}$$

where $\mathbf{k}$ is the comoving wave vector and $\mathbf{x}$ is the comoving space vector. Applying the Fourier decomposition to our Eq. (73) while the source is active yields

$$-k^2 \ddot{\tilde{\delta}}_k^{(2)}(t) + H(t)(-k^2)\dot{\tilde{\delta}}_k^{(2)}(t) - 2H^2(t)\left(1 - \frac{1}{6}\left(\frac{k}{a(t)H(t)}\right)^2\right)(-k^2)\tilde{\delta}_k^{(2)}(t)$$

$$= -\frac{8}{3}\frac{k^2}{a(t)^2}a(t)^2 \, 8\pi G \, \rho_{\mathrm{GW}}(k/a(t)). \tag{79}$$

and the factor $-k^2$ can be canceled such that

$$\ddot{\tilde{\delta}}_k^{(2)}(t) + H(t)\dot{\tilde{\delta}}_k^{(2)}(t) - 2H^2(t)\left(1 - \frac{1}{6}\left(\frac{k}{a(t)H(t)}\right)^2\right)\tilde{\delta}_k^{(2)}(t) = \frac{8}{3}8\pi G \, \rho_{\mathrm{GW}}(k/a(t), t). \tag{80}$$

For sub-horizon modes we can neglect the unity on the left hand side of the equation. Additionally, the right hand side can be formulated in terms of standard abundance $\Omega_{\mathrm{GW}}$ by replacing $\rho_{\mathrm{GW}} = \rho_{\mathrm{tot}}\Omega_{\mathrm{GW}}$ and using $\rho_{\mathrm{tot}} = \frac{3H_*^2}{8\pi G}$ at the time of the phase transition. We assume that the generation of GWs coincides with the duration of the FPT and thus completes within less than a Hubble time. Hence we can neglect the friction term $H(t)\dot{\tilde{\delta}}_k^{(2)}(t)$, approximate $a(t) \approx a(t_*) \approx a(t_* + 1/\beta)$ and $H(t) \approx H(t_*) \approx H(t_* + 1/\beta)$ and use $\rho_{\mathrm{GW}}$ from the previous subsection. After the phase transition completes the energy density of GWs simply redshifts as a radiation and we assume that during that time its power as source is negligible. In total our result reads

$$\ddot{\tilde{\delta}}_k^{(2)}(t) + \frac{1}{3}\frac{k^2 c^2}{a(t_*)^2}\tilde{\delta}_k^{(2)}(t) = 8H_*^2 \Omega_{\mathrm{GW}}(k/a(t_*), t) \quad \text{for } t \in [t_*, t_* + 1/\beta], \tag{81}$$

$$\ddot{\tilde{\delta}}_k^{(2)}(t) + H(t)\dot{\tilde{\delta}}_k^{(2)}(t) + \frac{1}{3}\frac{k^2 c^2}{a(t)^2}\tilde{\delta}_k^{(2)}(t) =$$

$$8H(t)^2\left(\frac{a_*}{a(t)}\right)^4 \Omega_{\mathrm{GW}}(k/a(t), t) \approx 0 \quad \text{for } t > t_* + 1/\beta. \tag{82}$$

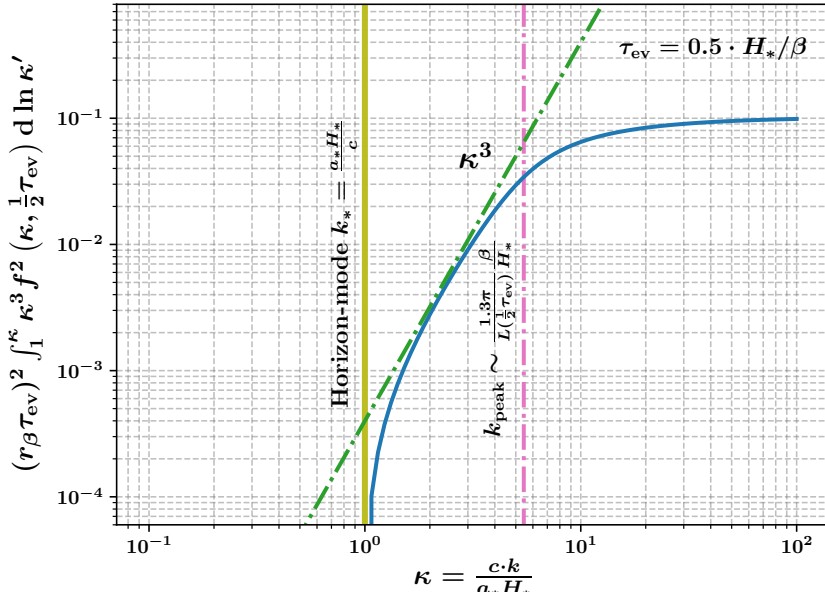

Figure 5: The integrated dimensionless power spectrum of GWs sourced by a FPT. The spectrum is evaluated in the middle of the phase transition $\tau_{ev}$. The yellow line shows the horizon mode and the pink dashed dotted line indicates the peak wave number of the logarithmic GW abundance. We chose $\beta/H_* = 1$ for demonstration reasons.

Note that the right hand side of Eq. (82) decays as $H(t)^2/a(t)^4$ and thus can be safely neglected. We have checked this approximation semi-analytically and found it to be consistent, see Appendix C. Also note that the choice of gauge is of negligible importance for sub-horizon modes.

**Solving the equation:** Next, we solve Eq. (81) to find $\tilde{\delta}_k^{(2)}(t)$ and estimate its impact on the matter power spectrum. This requires us to calculate the energy density in GWs from the fractional, logarithmic energy density $\Omega_{GW}^{log}$ in Eq. (63). Integrating the equation gives

$$\Omega_{GW}(k,t) := \frac{1}{\rho_{tot}}\rho_{GW}(k,t) = \kappa_{eff}^2\left(\frac{H_*}{\beta}\right)^2\left(\frac{\alpha}{1+\alpha}\right)^2\int_{k_{min}}^{k}\Delta(k'/\beta,t,v_w)\mathrm{d}\ln k', \qquad (83)$$

where we take for $k_{min} = \frac{a_* H_*}{c}$ the inverse size of the horizon at transition time and $v_w = c$ for the bubble wall velocity. The resulting energy density of GWs as a function of the wave number is shown in Fig. 5 for the dimensionless time $\tau := H_* t$ and wave number $\kappa := \frac{c}{a_* H_*}k$. With this scaling the differential equation becomes

$$\tilde{\delta}^{(2)\prime\prime}(\kappa,\tau) + \frac{1}{3}\kappa^2\tilde{\delta}^{(2)}(\kappa,\tau) = 8\cdot\Omega_{GW}(\kappa,\tau), \qquad (84)$$

where primes denote the derivative with respect to unit free time $\tau := H_* \cdot t$. The numerical solution at the end of the phase transition $\tilde{\delta}^{(2)}(\kappa,\tau_* + \frac{H_*}{\beta})$ is shown in Fig. 6 for initial conditions $\tilde{\delta}^{(2)}(\kappa,\tau_*) = \tilde{\delta}^{(2)\prime}(\kappa,\tau_*) = 0$. For an analytical resolution of Eq. (84) in various simplifying limits see appendix D. How to interpret this equation? From the expansion of the pressure to second order we see that for adiabatic perturbations and small changes in the sound speed on

sub-horizon scales, that

$$p^{(2)} = c_s^2 \rho^{(2)} + \sigma s^{(2)} + \frac{\partial c_s^2}{\partial \epsilon} \rho^{(1)2} + \frac{\partial c_s^2}{\partial s} s^{(1)2} + \frac{\partial \sigma}{\partial s} s^{(1)2} \approx c_s^2 \rho^{(2)}, \tag{85}$$

with $\sigma := (\partial p / \partial s)$ [54]. Therefore, as for linear perturbations, photon perturbations are characterized by $c_s^2 = 1/3$ and hence Eq. (84) describes the evolution of photon perturbations $\tilde{\delta}^{(2)} \equiv \tilde{\delta}_\gamma^{(2)}$. Comparing this equation with the wave equation for photon perturbations in the photon-baryon fluid before photon decoupling [55, 56]

$$\ddot{\delta}_\gamma + c_s^2 \frac{k^2}{a^2} \delta_\gamma = \frac{4}{3} 4\pi G \left( \rho_D^{(0)} \delta_D + \rho_B^{(0)} \delta_B + \rho_\gamma^{(0)} \delta_\gamma \right), \tag{86}$$

$$\delta_\gamma'' + c_s^2 \kappa^2 \delta_\gamma = 2 \left( \Omega_D^{(0)} \delta_D + \Omega_B^{(0)} \delta_B + \Omega_\gamma^{(0)} \delta_\gamma \right), \tag{87}$$

we notice, that what we found is a very similar system. But in our case their oscillations are driven by the gravitational wave density instead of matter or radiation density component. Since at that time baryons are still tightly coupled to photons they follow almost the same wave equation and thus we interpret our findings as baryon acoustic oscillations (BAOs) at a second order perturbative level driven by the GW energy density. As seen in Fig. 6 the oscillations lie on top of a dominant peak. The typical sound horizon of the oscillations is given by

$$r_s^{\text{GW}} := \int_{t_*}^{t_*+1/\beta} \frac{dt}{a_*} c_s = \frac{1}{\sqrt{3} a_* \beta} \frac{2\pi}{\sqrt{3} k_{\text{peak}}}. \tag{88}$$

For comparison, the typical sound horizon for standard BAOs and our BAOs is

$$r_s = 147 \, \text{Mpc} \, [57, 58], \tag{89}$$

$$r_s^{\text{GW}} = 3 \, \text{Mpc}, \tag{90}$$

respectively. Like for the standard BAOs after photon decoupling the baryons will transfer this information gravitationally to the dark matter perturbations and will thus be imprinted in the matter power spectrum.

Let us estimate the time on which a FPT has to occur in order to impact the matter power spectrum by density fluctuations produced via Eq. (84). The typical comoving scale on which the GW energy density per logarithmic frequency $\Omega_{\text{GW}}^{\log}(k, t)$ peaks at the end of the transition is at

$$\frac{k_{\text{peak}}}{a_*} \approx \frac{2\pi \beta}{c}, \tag{91}$$

with the phase transition duration $1/\beta$. Around this scale, the source term, the fractional energy density $\Omega_{\text{GW}}$, becomes approximately constant (see Fig. 5). Hence we can use it as a typical scale which will also be inherited to the induced density perturbations via Eq. (84). We rewrite the phase transition duration in terms of the Hubble parameter $\beta = r_\beta H_* = r_\beta H(t_*)$, where $r_\beta > 1$ for transitions shorter than a Hubble time. Therefore, the comoving wave number where the density fluctuation spectrum is approximately maximal is

$$k_{\text{peak}} = \frac{2\pi r_\beta H_* a_*}{c}, \quad \text{or}$$

$$\kappa_{\text{peak}} = 2\pi r_\beta. \tag{92}$$

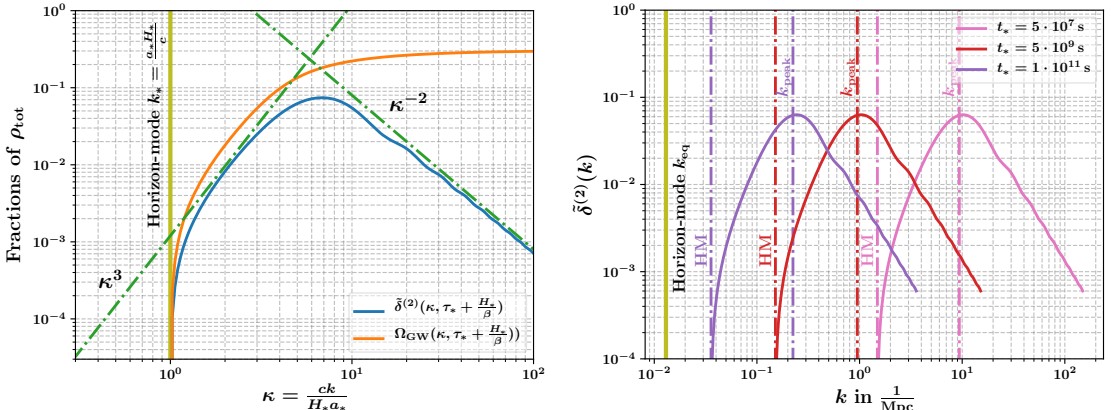

Figure 6: **Left:** In blue the numerical solution of Eq. (84) at the end of the phase transition and with initial conditions $\tilde{\delta}^{(2)}(\kappa, t) = \tilde{\delta}^{(2)\prime}(\kappa, t) = 0$ and for $\alpha \to \infty$ and $\beta/H_* = 1$. We also show the source term in orange at the end of the phase transition $\tau_{\mathrm{ev}} = \tau_* + H_*/\beta$. **Right:** Induced density perturbation by GWs from FPT. For demonstration purposes we chose $\alpha \to \infty, \beta = H_*$. Shown are solutions for different scales $a_* H_*$ in comparison with the horizon at matter-radiation equality.

This is analogous to a primordial density fluctuation which enters the horizon at $H_* a_*$, only that in our case we can shift the scale relative to $H_* a_*$ by the duration ratio $r_\beta$ of the phase transition.

From then on the scale of the density fluctuation is fixed and the time of a phase transition that impacts the matter power spectrum at its typical scale must fulfil the condition

$$\frac{2\pi r_\beta H_* a_*}{c} \gtrsim k_{\mathrm{eq}}. \tag{93}$$

This condition is met by *late phase transitions* around

$$t : 10^6 \, \mathrm{s} - t_{\mathrm{eq}} \sim T : (\mathcal{O}(100) - \mathcal{O}(1)) \, \mathrm{eV}. \tag{94}$$

We calculate the Hubble rate for these times using

$$H(t) = \frac{\dot{a}}{a} = H_0 \sqrt{\Omega_{m0}} \frac{\sqrt{a + a_{\mathrm{eq}}}}{a^2}, \tag{95}$$

where $H_0 \approx 70 \, \mathrm{Mpc}/(\mathrm{km\,s}) \approx 2.27 \cdot 10^{-18} \mathrm{s}^{-1}$ denotes the Hubble rate today and $a_{\mathrm{eq}} = \frac{\Omega_{\mathrm{rad}0}}{\Omega_{m0}} = \frac{8.5 \cdot 10^{-5}}{0.3} = 2.4 \cdot 10^{-4}$ is the scale factor at equality (the Hubble rate is $H_{\mathrm{eq}} = H(a_{\mathrm{eq}}) = 9.1 \cdot 10^{-14} \frac{1}{\mathrm{s}}$ or $t_{\mathrm{eq}} \approx 70000$ years). Integrating this equation leads to an implicit equation for the scale factor

$$t \cdot H_0 = \frac{2}{3} \frac{1}{\sqrt{\Omega_{m,0}}} \left[ \sqrt{a + a_{\mathrm{eq}}}(a - 2a_{\mathrm{eq}}) + 2a_{\mathrm{eq}}^{3/2} \right]. \tag{96}$$

For events sufficiently far enough from equality $a \ll a_{\mathrm{eq}}$ we can approximate Eq. (96) and get as limiting equation for the scale factor $a(t) = \sqrt{3 \cdot H_0 \sqrt{\Omega_{\mathrm{rad},0}} \cdot t}$.

**Impact on matter power spectrum:** Due to the production of extra deviations $\tilde{\delta}^{(2)}$ from the energy density by the phase transition the primordial modes around $k_* = 2\pi a_* H_*/c$ experience

a modification compared to their standard evolution. The change is captured by the transfer function

$$T^2_{\tilde{\delta}^{(2)}}(k) := 1 + \left( \frac{\tilde{\delta}^{(2)}(k)}{\delta^{(1)}_*(k)} \right)^2 , \tag{97}$$

where $\delta^{(1)}_*(k)$ are the primordial perturbations inside the horizon at $t_*$. Then the altered matter power spectrum with the amplitude at matter radiation equality compared to the spectrum today $\mathcal{P}_0(k)$ is

$$\tilde{\mathcal{P}}_{\text{eq}}(k) = T^2_{\tilde{\delta}^{(2)}}(k) \mathcal{P}_0(k) D^2_+(a_{\text{eq}}) , \tag{98}$$

with the approximate linear growth function

$$D_+(a) \approx \frac{5}{2} \frac{a\Omega_{m0}}{\Omega^{3/4}_{m0} - \Omega_\Lambda + (1 + \Omega_{m0}/2)(1 + \Omega_\Lambda/70)} , \tag{99}$$

which is $D_+(a_{\text{eq}}) \approx 2.5 \cdot 10^{-4}$ around equality. The linear matter power spectrum linearly extrapolated to today is given by the fitting formula [59]

$$\mathcal{P}_0(k) = A_0 \, k \cdot \frac{\ln(1 + c_1 q)}{c_1 q} \cdot \left( 1 + (c_2 q) + (c_3 q)^2 + (c_4 q)^3 + (c_5 q)^4 \right)^{-\frac{1}{4}} , \tag{100}$$

with

$$q := \frac{k}{\Omega_{m0} h \cdot \exp\left(-\Omega_{\text{baryon0}} - \sqrt{2h} \cdot \frac{\Omega_{\text{baryon0}}}{\Omega_{m0}}\right)} \approx 0.073 \frac{k}{k_{\text{eq}}} , \tag{101}$$

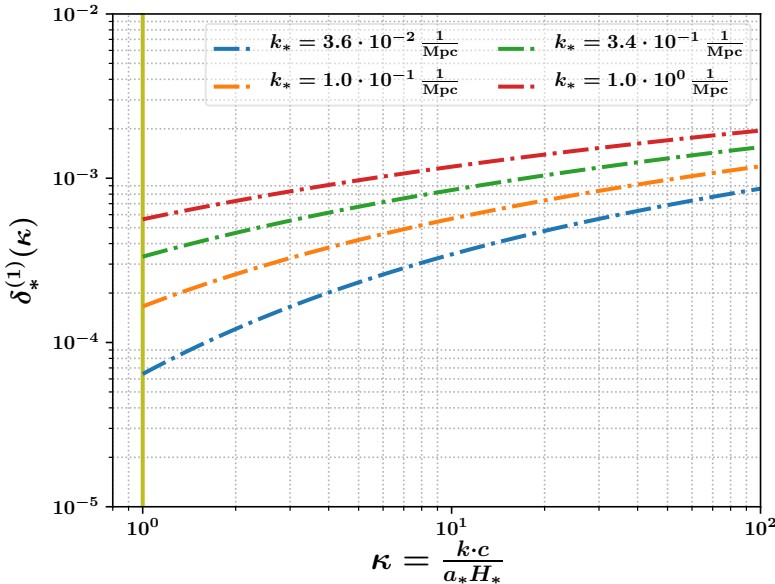

Figure 7: The estimated first order density fluctuations from the matter power spectrum in Eq. (103) as a function of dimensionless wave number for different times. Here $k_* = \frac{a_* H_*}{c}$. The $\kappa$-axis and the Hubble horizon shown in yellow are given in terms of the respective time for each of the curves.

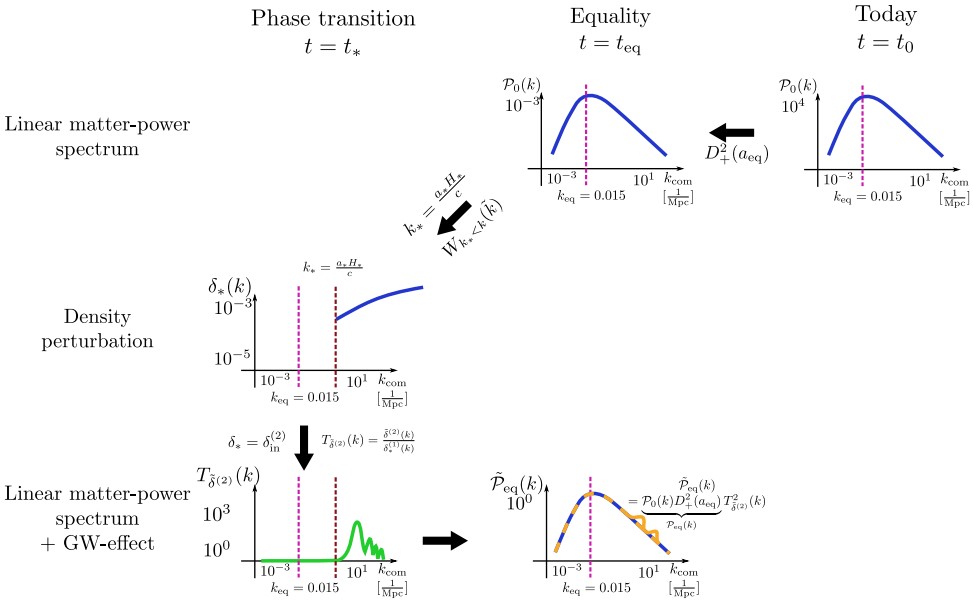

Figure 8: Schematic summary of the deduction of the matter power spectrum with the effect from gravitational wave induced second order density fluctuations.

and $c_1 = 2.34, c_2 = 3.89, c_3 = 16.1, c_4 = 5.46$ and $c_5 = 6.71$. The reduced Hubble parameter is set to $h = 0.7$ and the abundance of baryons today is $\Omega_{\text{baryon0}} = 0.05$ and $\Omega_{m0} = 0.3$ [58]. The amplitude $A_0$ of $\mathcal{P}_0(k)$ is calibrated such that the variance becomes [58]

$$0.8^2 = \sigma_8^2 = \int_0^\infty \mathrm{d}\tilde{k} \frac{\tilde{k}^2}{2\pi^2} \mathcal{P}_0(\tilde{k}) \times \left( \frac{3 j_1(\tilde{k}R)}{(\tilde{k}R)} \right)^2. \tag{102}$$

Here $j_1(x) := \sin(x)/x^2 - \cos(x)/x$ and $R = 8 \, \text{Mpc}/h$.

In order to derive the transfer function $T_{\tilde{\delta}^{(2)}}(k)$ we estimate the amplitude of a typical density perturbation $\delta_*^{(1)}(k)$ for sub-horizon modes as standard deviation from $\mathcal{P}$

$$\delta_*^{(1)}(k) \cong D_+(a_{\text{eq}}) \sqrt{\int_0^\infty \mathrm{d}\tilde{k} \frac{\tilde{k}^2}{2\pi^2} \mathcal{P}_0(\tilde{k}) W_k^2(\tilde{k})} \quad \text{for} \quad k_* \leq k, \tag{103}$$

where $W_k(\tilde{k}) = \frac{3}{(\tilde{k}/k)^3}(\sin(\tilde{k}/k) - \tilde{k}/k \cos(\tilde{k}/k))$ is called window function. The restriction to modes with $k_* < k$ is necessary since only modes that have entered the horizon at the time of the phase transition are relevant. In Fig. 7 we show Eq. (103) in terms of the dimensionless wave number $\kappa$, $\delta_*^{(1)}(\kappa \cdot k_*)$ for $10^0 \leq \kappa$, at different transition times.

We use the estimated primordial density fluctuations to define the transfer function Eq. (97) which is show in Fig. 9 for some example cases together with the modified matter power spectrum $\tilde{\mathcal{P}}$ at equality. Note that $\delta^{(2)} \ll 1$ is fulfilled at all times, see for example Fig. 7. The whole procedure is schematically summarized in Fig. 8.

As seen in Figs. 9, 10 and 11 the GWs produced by the FPT imprint a peak on the matter power spectrum around the comoving scale $k_{\text{peak}}$. The transfer functions decrease rapidly with smaller phase transition duration $r_\beta$ and also with smaller strength $\alpha$. This behaviour is expected from the prefactors of the GW energy density in Eq. (83).

Hence, the height of the peak is determined by the parameters $t_*$, $\alpha$ and $r_\beta$. We can put limits on them by requiring that the height of the peak should not exceed the bound set by the cosmic

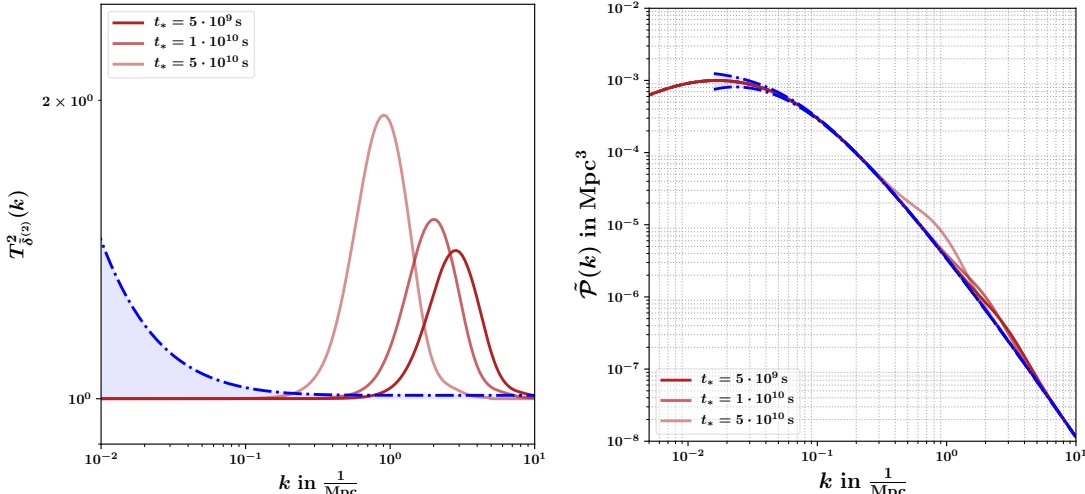

Figure 9: The impact of a late phase transition for different starting times $t_*$ on the linear matter power spectrum. We chose for each time an inverse duration of $\beta = 3H_*$ and latent heat $\alpha = 3$, respectively.

variance of the linear matter power spectrum. The latter is defined via [60]

$$\sigma(k) := \sqrt{\text{cov}(\mathcal{P}_0(k), \mathcal{P}_0(k))} = \mathcal{P}_0(k)\sqrt{\left(\frac{2}{N} + \frac{1}{n}\right)}, \tag{104}$$

which holds for Gaussian random fields. $N$ denotes the number of modes and $n$ is related to the so called band-averaged trispectrum which can be estimated to $1/n \approx 0.0079^2 \, (\text{Gpc}/h)^3/V$ [61]. We estimate the number of modes as $2/N = (2\pi)^2/(V \cdot k^2 \Delta k)$ with $V \approx 1(\text{Gpc}/h)^3$ and $\Delta k \sim 0.02 \cdot \log(k\,\text{Mpc})(\text{Mpc}/h)^{-1}$ (typical distance between galaxies) which reproduces approximately the cosmic variance found in [61].

The modified matter power spectrum $\tilde{\mathcal{P}}$ should not exceed this bound, i.e

$$\tilde{\mathcal{P}}(k) \ll \sigma(k), \quad \forall k. \tag{105}$$

In Fig. 12 we show a parameter scan in the $\alpha$-$\beta$-plane for different phase transition times $t_*$. The red shaded regions are excluded by the cosmic variance bound, while values in the white region are consistent with it. We observe that only very long $r_\beta < 5 - 6.8$ and strong $\alpha > 1$ phase transitions can be ruled out.

The earlier the phase transition takes place the less is it constrained. FPTs with such extreme parameter values have been proposed in the past. Long lasting transitions are realized for example in SUSY [62] and models with a lot of supercooling are for example Randall-Sundrum, composite Higgs models [42] and models with an almost conformal symmetry in general [16]. In Fig. 13 we convert contour line values into a bound on the GW signal today in the standard frequency - GW abundance plane. The logarithmic GW abundance today due to bubble collisions is [34]

$$h^2\Omega_{\text{GW}0}^{\log}(f) = 1.67 \times 10^{-5} \cdot r_\beta^{-2} \cdot \left(\frac{\kappa_{\text{eff}}(\alpha)\alpha}{1+\alpha}\right)^2 \left(\frac{100}{g_*}\right)^{\frac{1}{3}} \left(\frac{0.11 v_w^3}{0.42 + v_w^2}\right) \frac{3.8(f f_{\text{peak}})^{2.8}}{1 + 2.8(f/f_{\text{peak}})^{3.8}}, \tag{106}$$

with the peak frequency $f_{\text{peak}}$ today

$$f_{\text{peak}} = 16.5 \times 10^{-6} \, \text{Hz} \frac{0.62}{v_w^2 - 0.1 v_w + 1.8} r_\beta \left(\frac{T_*}{100}\right)\left(\frac{g_*}{100}\right)^{\frac{1}{6}}. \tag{107}$$

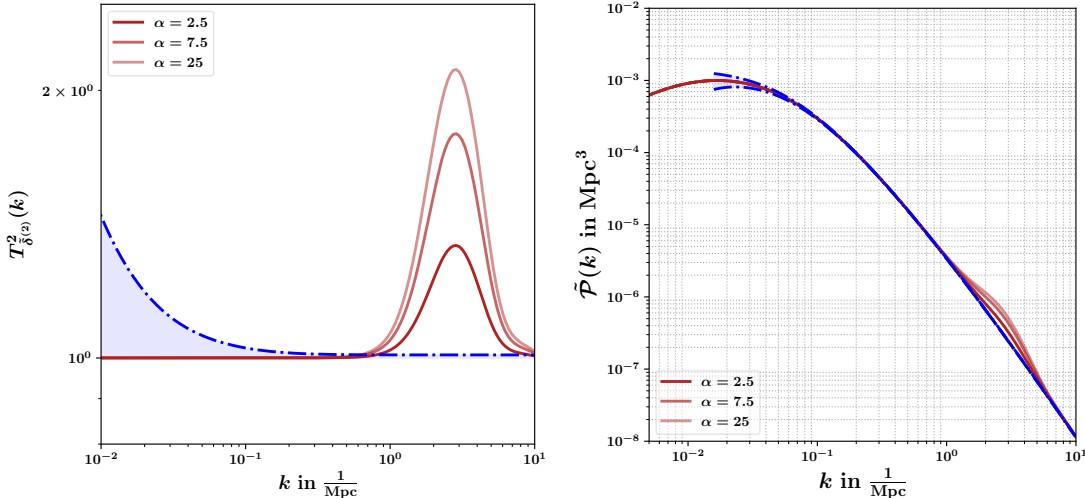

Figure 10: The impact of a late phase transition for different strength $\alpha$ on the linear matter power spectrum. We chose for each $\alpha$ an inverse duration of $\beta = 3H_*$ and fixed the transition time to $t_* = 5 \times 10^9$ s, respectively.

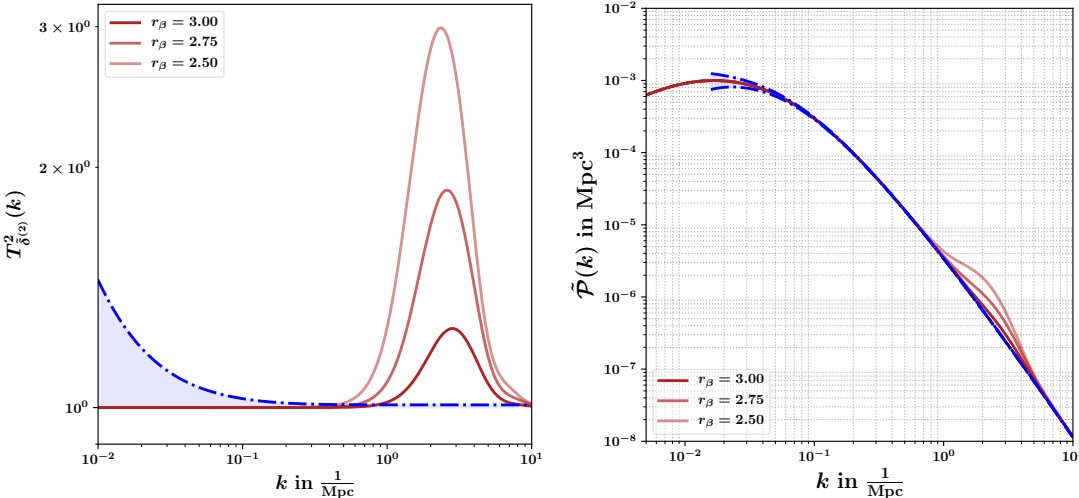

Figure 11: The impact of a late phase transition for different duration ratios $r_\beta$ on the linear matter power spectrum. We chose for each $r_\beta$ a strength $\alpha = 3$ and fixed the transition time to $t_* = 5 \times 10^9$ s, respectively.

The time of the phase transition can be converted into the temperature of the plasma by

$$T_* = \frac{30}{\pi^2} \frac{3}{4g_*^3} \left( \frac{1}{8\pi G t_*^2} \right)^{\frac{1}{4}} \tag{108}$$

and the number of relativistic degrees of freedom $g_*$ after the QCD phase transition (and hence for a late phase transition) is 3.36. Note, that for BSM models this value differs depending on the field content and their properties. The frequency window is set to $f \approx 1.5 \times 10^{-16}$ Hz as lower bound and $f \approx 1.5 \times 10^{-14}$ Hz as upper bound which approximately corresponds to the right, big $k$ slope of the matter power spectrum. For smaller frequencies our assumption of a radiation dominated universe becomes very weak.

For comparison, we show also the projected bounds for LISA [2], the timing pulsar arrays [63] NANOGrav [64,65], PPTA [66], EPTA [67] and CMB [68–70].

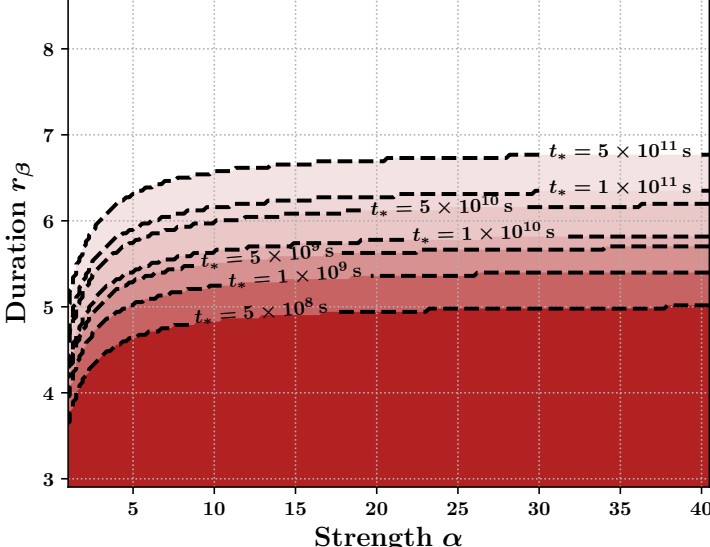

Figure 12: Parameter scan of $\alpha$ and $r_\beta$ for different transition times $t_*$. The red colored regions are excluded by cosmic variance and thus a late FPT with these parameter combination would change structure formation too strongly.

Following reference [17,71,72] the GW energy density can be limited by the effective number of neutrino species $N_\nu$ via

$$h^2\Omega_{\text{GW}}(f) \leq 5.6 \cdot 10^{-6}\Delta N_\nu, \tag{109}$$

where $\Delta N_\nu$ denotes the deviation from the SM value $N_\nu = 3$. BBN constrains this number to $\Delta N_\nu \leq 0.2$ [73] giving the bound on the allowed amount of GW before BBN shown in Fig. 13. The indirect bound for the CMB is taken from [68].

## 5 Conclusion

Let us summarize the results. In this work we have studied the possible impact of a FPT on small scale structure via the production of GWs in the radiation dominated epoch. A linear relation between the energy density of GWs $\rho_{\text{GW}}(k)$ and adiabatic density perturbations has been found by expanding the full non-linear Eqs. (5) and (6) to second order in the $1 + 3$ covariant formulation. In this formalism the spacetime is decomposed into the direction of fluid flow and its orthogonal hyper-surface. Then, a set of gauge invariants to first order with clear geometrical interpretations can be constructed.

When only considering parameters for which the GW energy density surpasses the other source terms during the transition, the adiabatic density perturbations follow a wave equation which is driven by the GW energy density. In this case our equation describes photon acoustic oscillations induced by the GW energy density. Since the photons are still coupled to the baryons at such times, the baryons undergo the same oscillations which manifest themselves eventually in the matter power spectrum.

Since phase transitions are typically taking place within the Hubble horizon $H_*$ at the time of the transition the scale on which the perturbations are affected is bounded by the horizon size $k_* = a_* H_* / c$. However, we found that the scale that is maximally impacted equals the scale where the GW energy density per logarithmic frequency has a maximum $k_* = 2\pi r_\beta a_* H_* / c$. This implies that the linear matter power spectrum, if at all, can only be affected on the length

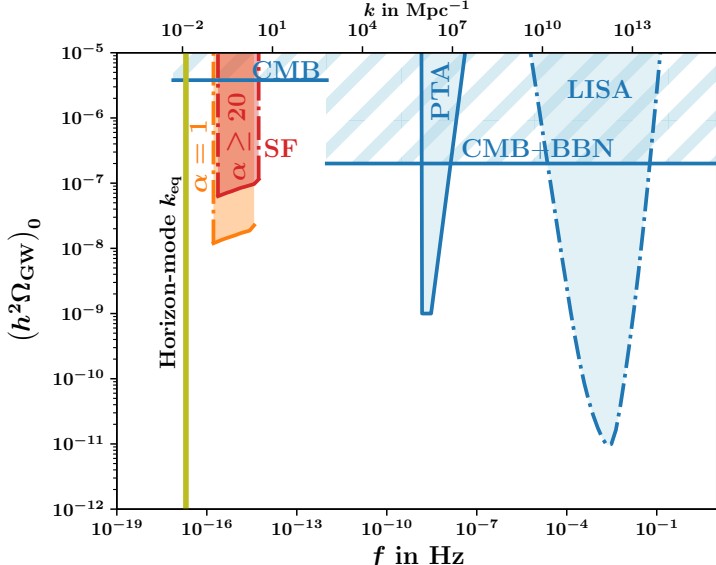

Figure 13: The GW abundance today as a function of today frequency. Shown are the projected bound by LISA [2] (dashed-dot blue), the bound set by PTAs [64–67] (solid blue) and the indirect bounds from CMB and BBN [68–70] (hatched areas). As the result of our calculation, the red colored regions are excluded by structure formation. However, many assumptions went into our calculation so the bound should be taken with care.

scales of galaxies and above if the transition occurred at very late times $\geq 10^6$ s but still within the radiation dominated regime. Late phase transitions and their impact on structure formation (also due to gravitational waves) have been discussed in the past, for example in the matter dominated era [74–76] (in the literature the phrase late time phase transition is sometimes used for transitions after equality or photon decoupling). Specific particle models have been discussed in [77] and a model with a very late phase transition including a dark energy component is presented in [78].

The maximally allowed duration $\beta^{-1}$ and strength $\alpha$ of a phase transition is bounded by cosmic variance and depends on the time of the transition. We find that this bound constrains these parameters only very weakly, excluding transitions that last longer than $\gtrsim 1/6.8$ Hubble times in the case the transition is close to equality and $\gtrsim 1/5$ in the case the transition takes place on galaxy scales. From the parameter set $t_*, \beta$ and $\alpha$ we derived the GW abundance per logarithmic frequency today and translated the bounds from structure formation into an exclusion region in Fig. 13.

Our results are based on the following assumptions. First of all we looked at adiabatic perturbations only. We simplified our calculation further by neglecting anisotropy and vorticity effects as well as current density effects. In principle the anisotropic stress could be also have effects on the matter power spectrum directly. As a next step it would be reasonable to study the possible effects of the anisotropic stress on the density perturbations in more details. For example, its scalar part (corresponding to the quadruple term in the momentum distribution caused by the bubble collision in the fluid) could constitute a difference in the Bardeen potentials $\Psi - \Phi \sim \Pi$ analogous to neutrino and photon anisotropies and in this way even affect linear perturbations. The effect of an extra anisotropic stress on the CMB and on curvature perturbations has been discussed in [79] also using the $1+3$ covariant formalism. Additionally, one could consider effects of the anisotropic stress on a second perturbative level. A non zero and transverse anisotropic stress tensor can appear in the non-linear Eqs. (5), (6) and

the conservation laws Eqs. (A.20), (A.21) coupled to the acceleration $A_a$ and the shear $\sigma_{ab}$. The acceleration $A_a^{(2)}$ is thus not parallel to the density gradient any more which will make the calculation much more complex when including the anisotropic stress.

In our derivation we assumed the equation of state parameter $\omega$ and the sound speed $c_s^2$ are constant in time and space and also that $\frac{\delta\rho}{\delta p} = \frac{\delta^2\rho}{\delta^2 p} = c_s^2$. In general these parameters could depend on space and time. However, on the one hand the decline of $\omega$ close to equality is very gentle and on the other hand the change within a Hubble time is expected to be negligible. As closer we get to matter-radiation equality $\omega$ departs more and more from being 1/3. A rough estimation gives $\omega \approx 0.27$ at $t \approx 10^{11}$ s.

In this work we found that only strong GWs sourced by phase transitions with a lot of super-cooling can have effects on structure. In this regime the bubble dynamics is fixed to bubbles expanding into vacuum and hence the only source of GWs are bubble collisions.

Note also that our study is limited to phase transitions on sub-horizon scales which complete within a Hubble time $\beta > H_*$. Our results are very close to this boundary and hence effects of the Hubble friction terms in the wave equation for the GWs and the density perturbations might suppress the amplitudes even further, shrinking the constrained region in parameter space.

For perturbations induced from phase transitions at high wavenumbers, it should be also mentioned that these will experience non-linear growth at later times. Hence, if there was an impact at those scales, it will be overlaid by non-linear structure formation.

In future work we will look at more direct consequences of the phase transition on structure formation. One idea is to take up on the work done Schmid et al. [25, 26] and study the effect on linear perturbations by changing sound speed. As mentioned, in [27] it was shown that the sound speed does not change a lot in particle models with many scalar fields, but could depart from $1/\sqrt{3}$ in fermion rich models. Another possibility is to study the direct impact of the anisotropic stress on linear perturbations, as mentioned before through the difference in the Bardeen potentials $\Psi - \Phi \sim \Pi$. Also, the huge amount of supercooling $\alpha \gg 1$ in our calculation turns the background cosmology from radiation dominated to vacuum energy dominated such that the equation of motion of the linear density perturbations changes which could also lead to direct effect on the matter power spectrum.

## Acknowledgements

The authors thank Ruth Durrer for advice in the initial stages of this project and for helpful comments on the final version of the paper. CD would like to acknowledge insightful discussions with Andreas Trautner. S.C.C. would like to thank the Max-Planck-Institut für Kernphysik in Heidelberg for hospitality during his visit, where this work was initiated. The work of S.C.C. is supported by the Spanish grants SEV-2014-0398, FPA2017-85216-P (AEI/FEDER, UE), PROM-ETEO/2018/165 (Generalitat Valenciana) and BES-2016-076643. This work is supported by the Deutsche Forschungsgemeinschaft (DFG, German Research Foundation) under Germany's Excellence Strategy EXC 2181/1 - 390900948 (the Heidelberg STRUCTURES Excellence Cluster).

## A  The 1+3-covariant formulation

Quantities in perturbation theory are in general gauge dependent, i.e. they change under infinitesimal coordinate transformations $\tilde{x}^\mu = x^\mu + \epsilon\,\xi^\mu$, where $\epsilon \ll 1$ is a small parameter and $\xi^\mu \in \mathbb{R}^4$ is some vector field. Under a linear perturbation an arbitrary tensor field $S$ is

split into its zeroth-order part[3] $S^{(0)}$, also referred to as background value, and its first-order part $\delta S \equiv \epsilon S^{(1)}$, i.e. $S = S^{(0)} + \epsilon S^{(1)}$. Under a gauge transformation the latter perturbative component is not simply mapped to itself but rather receives an additional term dependent of the gauge vector $\xi^\mu$ according to

$$S^{(1)} \to S^{(1)} + \mathcal{L}_\xi S^{(0)}, \tag{A.1}$$

where this additional term is the Lie-derivative $\mathcal{L}_\xi$ along $\xi^\mu$ of the background term $S^{(0)}$ [52, 80, p.59]. In order to make universally valid predictions for a perturbative physical model we need to introduce gauge invariant quantities.

A tensor field is called gauge invariant to first order if for any vector field $\xi^\mu$ the Lie-derivative vanishes $\mathcal{L}_\xi S^{(0)}$. Based on the gauge transformation rule in Eq. (A.1) the Stewart & Walker lemma [81] states that a tensor is gauge invariant[4] if and only if it either vanishes in the background, is a constant scalar on the background or can be written as a sum of products of Kronecker-deltas with constant coefficients [52].

In the spirit of this lemma, Ellis, Bruni and co-authors developed the so called $1 + 3$ covariant formulation of gravity [22,23] based on earlier papers by Heckmann and Schücking [18], Raychaudhuri [19], Ehlers [20] and Hawking [21]. In this section we will follow closely ref. [28]. The advantage of this approach resides in the simple geometric meaning of the central variables and their gauge invariance which is due to the fact that they vanish in a spatially homogeneous background, for example in the background of a Friedmann-Lemaître-Robertson-Walker (FLRW) metric. These variables are constructed by decomposing the spacetime into the direction of the four-velocity of a comoving observer that follows the fluid flow lines $x^a$ and the projection tensor into the instantaneous rest space of $u^a$,

$$u^a = \frac{\mathrm{d}x^a}{\mathrm{d}\tau} \quad \text{and} \quad h_{ab} := g_{ab} + u_a u_b, \tag{A.2}$$

with the proper time $\tau$, $u^a u_a = -1$ and $g_{ab}$ being the metric tensor with signature $(-+++)$. We follow the convention of the literature and use Latin indices for four-vectors $a, b, c, \cdots = 0, 1, 2, 3$ and $\alpha, \beta, \gamma \cdots = 1, 2, 3$ for spacelike three-vectors.

The two tensors are perpendicular projectors

$$h_{ab} u^b = g_{ab} u^b + u_a u_b u^b = u_a - u_a = 0 \tag{A.3}$$

that project a spacetime quantity onto the flow lines or in the orthogonal direction which enables a unique splitting into irreducible timelike and spacelike components (establishing the name $1 + 3$).

Exemplarily the time- and space derivative of a general tensor $S_{ab\ldots}{}^{cd\ldots}$ is obtained by projecting the covariant derivative $\nabla_a$:

$$\dot{S}_{ab\ldots}{}^{cd\ldots} := u^e \nabla_e S_{ab\ldots}{}^{cd\ldots} \quad \text{and} \quad \mathrm{D}_e S_{ab\ldots}{}^{cd\ldots} := h_e{}^s h_a{}^f h_q{}^c \cdots \nabla_s S_{f\ldots}{}^{q\ldots}. \tag{A.4}$$

The next step is to describe the kinematics of an observer in this framework under the influence of gravity and matter represented by the energy momentum tensor $T_{ab}$. We will set the speed of light $c = 1$ and the gravitational coupling $\kappa := 8\pi G = 1$, where $G$ is the gravitational constant.

---

[3]Subscripts in parentheses denote the perturbative order.

[4]Unless stated otherwise, we mean by gauge invariant always gauge invariant to linear order.

## A.1 Kinematic variables

In the 1+3-covariant approach to gravity, the kinematic quantities that determine the motion of a test particle are the tracefree *shear* tensor $\sigma_{ab} := D_{\langle b}u_{a\rangle}$, the antisymmetric (hence tracefree) *vorticity* tensor $\omega_{ab} := D_{[b}u_{a]}$, the *volume expansion* scalar $\Theta := D^a u_a$ and the four-*acceleration* $A_a = u^b \nabla_b u_a$, which emerge from the irreducible decomposition of the covariant derivative of the four-velocity

$$\nabla_b u_a = \sigma_{ab} + \omega_{ab} + \frac{1}{3}\Theta h_{ab} - A_a u_b \,. \tag{A.5}$$

The here used brackets are defined as

$$S_{(ab)} := \frac{1}{2}(S_{ab} + S_{ba})\,, \qquad\qquad S_{[ab]} := \frac{1}{2}(S_{ab} - S_{ba})\,, \tag{A.6}$$

$$S_{\langle ab\rangle} := h_a^{\,c}h_b^{\,d}S_{cd} - \frac{1}{3}h^{cd}S_{cd}h_{ab}\,, \qquad V_{\langle a\rangle} := h_a^{\,b}V_b\,. \tag{A.7}$$

Useful identities for these objects are collected in Appendix A.7. In an FLRW universe at zeroth-order the shear, the vorticity and the acceleration vanish and hence the Stewart & Walker Lemma provides gauge invariant quantities [82]. Moreover in a spatially homogeneous model like the FLRW metric the background value of the scalar $\Theta^{(0)}(t)$ depends solely on time and thus the spatial derivative $D_a\Theta$ is equally gauge-invariant.

## A.2 Gravity

In general relativity gravitation arises from intrinsic properties of the spacetime manifold and matter. Einstein's field equations formulate this relation by

$$R_{ab} - \frac{1}{2}R g_{ab} = \kappa T_{ab} - \Lambda g_{ab}\,, \tag{A.8}$$

where $R_{ab}$ and $R$ are the Ricci tensor and scalar, respectively, and $\Lambda$ is the cosmological constant. The Ricci tensor is the contraction of the Riemann tensor $R_{abcd}$ which encodes the curvature of the spacetime manifold. The latter can be split into two parts

$$R_{abcd} = C_{abcd} + \frac{1}{2}(g_{ac}R_{bd} + g_{bd}R_{ac} - g_{bc}R_{ad} - g_{ad}R_{bc}) - \frac{1}{6}R(g_{ac}g_{bd} - g_{ad}g_{bc})\,. \tag{A.9}$$

While Ricci tensor $R_{ab}$ and scalar $R$ express volume changes due to a local matter source and hence reflect the local part of the gravitational field, the Weyl tensor $C_{abcd}$[5] contains information about the propagating degrees of freedom. Using the four-velocity vector, $C_{abcd}$ can be decomposed further into the so called electric and magnetic parts [83, 84]

$$E_{ab} = C_{acbd}u^c u^d \quad\text{and}\quad H_{ab} = \frac{1}{2}\epsilon_a^{\,cd}C_{cdbe}u^e, \text{ respectively}\,. \tag{A.10}$$

Both tensors are symmetric, tracefree and gauge invariant due to $C^{(0)}_{abcd} = 0$ in the FLRW background. As we shall see, the propagation of GWs is mainly governed by the magnetic part $H_{ab}$ while the electric part $E_{ab}$ is closely related to tidal forces.

Having discussed long range gravitational effects, let us focus on local gravity which is expressed by the Ricci tensor and the energy-momentum tensor. For a general fluid the energy-momentum tensor decomposes with respect to the fundamental timelike velocity field into

$$T_{ab} = \rho u_a u_b + 2u_{(a}q_{b)} + p h_{ab} + \pi_{ab}\,, \tag{A.11}$$

---

[5] The Weyl tensor shares all symmetries with the Riemann tensor $R_{abcd} = R_{cdcb}$, $R_{abcd} = R_{[ab][cd]}$ and $R_{a[bcd]} = 0$ and is, by construction, additionally tracefree.

where $\rho := T^{ab}u_a u_b$ is the *energy density*, $q_b := h_a{}^b T_{bc}u^c$ is the *energy current density*, $p := T_{ab}h^{ab}/3$ is the *pressure* and $\pi_{ab} := T_{\langle ab\rangle}$ the trace-free *anisotropic stress*. While both the anisotropic stress and the energy current density again vanish in a FLRW universe and are thus gauge invariant, the pressure and the energy density depend only on time and hence their spatial gradients are also gauge invariant to first order.

Rewriting Einstein's equation as $R_{ab} = \kappa(T_{ab} - \frac{1}{2}T_{ab}) + \Lambda g_{ab}$ leads to three equations that relate the Ricci tensor and the matter-fields [28],

$$R_{ab}u^a u^b = \kappa \frac{1}{2}(\rho + 3p) - \Lambda, \tag{A.12}$$

$$h_a{}^b R_{bc}u^c = -\kappa q_a, \quad \text{and} \tag{A.13}$$

$$h_a{}^c h_b{}^d R_{cd} = \kappa \frac{1}{2}(\rho - p)h_{ab} + \kappa \pi_{ab} + \Lambda h_{ab}. \tag{A.14}$$

## A.3 Equations of motion

As discussed in the previous sections, the $1+3$ covariant approach identifies gauge invariant components of the energy-momentum tensor, the Riemann tensor and the four velocity gradient with a clear geometrical and physical meaning. The equations of motion for these variables are inferred from the *Bianchi* and *Ricci* identities and are accompanied by constraint equations. The equations quoted in this section have been derived in [85, 86], for details see also the review [28]. Using Eq. (A.14) and the Bianchi identities for the Weyl tensor

$$\nabla^d C_{abcd} = \nabla_{[b}R_{a]c} + \frac{1}{6}g_{c[b}\nabla_{a]}R \tag{A.15}$$

one finds the non-linear propagation equations of the electric and magnetic components of the Weyl-tensor,

$$
\begin{aligned}
\dot{E}_{\langle ab\rangle} =\ & -\Theta E_{ab} - \frac{1}{2}\kappa(\rho+p)\sigma_{ab} + \operatorname{curl}H_{ab} - \frac{1}{2}\kappa\dot{\pi}_{ab} - \frac{1}{6}\kappa\Theta\pi_{ab} - \frac{1}{2}\kappa\operatorname{D}_{\langle a}q_{b\rangle} - \kappa A_{\langle a}q_{b\rangle} \\
& + 3\sigma_{\langle a}{}^c\left(E_{b\rangle c} - \frac{1}{6}\kappa\pi_{b\rangle c}\right) + \varepsilon_{cd\langle a}\left[2A^c H_{b\rangle}{}^d - \omega^c\left(E_{b\rangle}{}^d + \frac{1}{2}\kappa\pi_{b\rangle}{}^d\right)\right], \tag{A.16}
\end{aligned}
$$

$$
\begin{aligned}
\dot{H}_{\langle ab\rangle} =\ & -\Theta H_{ab} - \operatorname{curl}E_{ab} + \frac{1}{2}\kappa\operatorname{curl}\pi_{ab} + 3\sigma_{\langle a}{}^c H_{b\rangle c} - \frac{3}{2}\kappa\omega_{\langle a}q_{b\rangle} \\
& - \varepsilon_{cd\langle a}\left(2A^c E_{b\rangle}{}^d - \frac{1}{2}\kappa\sigma^c{}_{b\rangle}q^d + \omega^c H_{b\rangle}{}^d\right), \tag{A.17}
\end{aligned}
$$

while the spacelike constraints become

$$\operatorname{D}^b E_{ab} = \kappa\left[\frac{1}{3}\operatorname{D}_a\rho - \frac{1}{2}\operatorname{D}^b\pi_{ab} - \frac{1}{3}\Theta q_a + \frac{1}{2}\sigma_{ab}q^b\right] - 3H_{ab}\omega^b + \varepsilon_{abc}\left(\sigma^b{}_d H^{cd} - \frac{3}{2}\kappa\omega^b q^c\right) \tag{A.18}$$

and

$$\operatorname{D}^b H_{ab} = \kappa(\rho+p)\omega_a - \frac{1}{2}\kappa\operatorname{curl}q_a + 3E_{ab}\omega^b - \frac{1}{2}\kappa\pi_{ab}\omega^b - \varepsilon_{abc}\sigma^b{}_d\left(E^{cd} + \frac{1}{2}\kappa\pi^{cd}\right). \tag{A.19}$$

Here, the vorticity vector $\omega_a := \epsilon_{abc}\omega^{bc}/2$ has been introduced together with the projection $\epsilon_{abc} := \eta_{abcd}u^d$ of the totally antisymmetric tensor $\eta_{abcd}$.

Table 1: Perturbative expansion of the central quantities in the $1+3$ covariant theory in a FLRW background and their first order gauge invariant version. In the third column the first term refers to the zeroth-order component and the second term to $\delta S \equiv \epsilon S^{(1)}$. Since for most quantities the zeroth-order part is zero and hence $S = \delta S$ we often neither use the superposed index nor the $\delta$-symbol for their first-order term.

| Variable | Symbol | Perturbative Expansion $S = S^{(0)} + \epsilon S^{(1)}$ | First order GI |
|---|---|---|---|
| Energy density | $\rho$ | $\rho(t) + \rho(\mathbf{x}, t)$ | $D_a \rho(\mathbf{x}, t)$ |
| Pressure | $p$ | $p(t) + p(\mathbf{x}, t)$ | $D_a p(\mathbf{x}, t)$ |
| Anisotropic stress | $\pi$ | $0 + \pi(\mathbf{x}, t)$ | $\pi(\mathbf{x}, t)$ |
| Energy density current | $q$ | $0 + q(\mathbf{x}, t)$ | $q(\mathbf{x}, t)$ |
| Volume expansion | $\Theta$ | $\Theta(t) + \Theta(\mathbf{x}, t)$ | $D_a \Theta(\mathbf{x}, t)$ |
| Shear | $\sigma$ | $0 + \sigma(\mathbf{x}, t)$ | $\sigma(\mathbf{x}, t)$ |
| Vorticity | $\omega$ | $0 + \omega(\mathbf{x}, t)$ | $\omega(\mathbf{x}, t)$ |
| Acceleration | $A$ | $0 + A(\mathbf{x}, t)$ | $A(\mathbf{x}, t)$ |
| Long range grav. field (Weyl tensor) | $C$ | $0 + C_{abcd}(\mathbf{x}, t)$ | $C_{abcd}(\mathbf{x}, t)$ |

From the Bianchi identity expressing the conservation of energy and momentum $\nabla^a T_{ab} = 0$ we find the propagation equations for the energy density and the energy flux,

$$\dot{\rho} = -\Theta(\rho + p) - D^a q_a - 2A^a q_a - \sigma^{ab}\pi_{ab}, \tag{A.20}$$

$$\dot{q}_{\langle a \rangle} = -D_a p - (\rho + p)A_a - \frac{4}{3}\Theta q_a - (\sigma_{ab} + \omega_{ab})q^b - D^b \pi_{ab} - \pi_{ab}A^b. \tag{A.21}$$

Finally the Ricci-identities $2\nabla_{[a}\nabla_{b]}u_c = R_{abcd}u^d$ give the Raychaudhuri equation for the volume expansion and the propagation equations for the shear and the vorticity

$$\dot{\Theta} = -\frac{1}{3}\Theta^2 - \frac{1}{2}\kappa(\rho + 3p) - 2(\sigma^2 - \omega^2) + D^a A_a + A_a A^a + \Lambda, \tag{A.22}$$

$$\dot{\sigma}_{\langle ab \rangle} = -\frac{2}{3}\Theta\sigma_{ab} - \sigma_{c\langle a}\sigma^c{}_{b\rangle} - \omega_{\langle a}\omega_{b\rangle} + D_{\langle a}A_{b\rangle} + A_{\langle a}A_{b\rangle} - E_{ab} + \frac{1}{2}\kappa\pi_{ab}, \tag{A.23}$$

$$\dot{\omega}_{\langle a \rangle} = -\frac{2}{3}\Theta\omega_a - \frac{1}{2}\operatorname{curl}A_a + \sigma_{ab}\omega^b. \tag{A.24}$$

These identities also imply the following constraints for the shear, the vorticity and the magnetic component of the Weyl tensor

$$D^b \sigma_{ab} = \frac{2}{3}D_a\Theta + \operatorname{curl}\omega_a + 2\varepsilon_{abc}A^b\omega^c - \kappa q_a, \quad D^a\omega_a = A_a\omega^a, \tag{A.25}$$

$$H_{ab} = \operatorname{curl}\sigma_{ab} + D_{\langle a}\omega_{b\rangle} + 2A_{\langle a}\omega_{b\rangle}. \tag{A.26}$$

These equations allow us to find the behavior of density perturbations and GWs on a FLRW background. For a detailed derivation of these equations see [28]. In table 1 we have summarized the central quantities of the $1+3$ approach, their interpretation and their gauge properties.

### A.4 Linear density perturbations

Spatial inhomogeneities in the matter density are described by the spatial comoving fractional gradient and the comoving expansion gradient [22]

$$\Delta_a := \frac{a}{\rho} D_a \rho, \tag{A.27}$$

$$Z_a := a D_a \Theta, \tag{A.28}$$

which are both orthogonal to the fluid flow. In a spatially homogeneous background they are gauge invariant because $\rho$ and $\Theta$ depend only on time such that the spatial gradient $D_a \rho|_{\text{background}} = D_a \rho(t) = 0$ vanishes in the background. The time- and space dependent variations of over- and under densities which are expressed by the orthogonal projected divergence of the comoving fractional gradient $a D^a \Delta_a =: \Delta$ are closely related to the Laplacian of the density contrast $\delta := \delta\rho/\rho$. However, besides the usual over- and under densities also distortions $\Delta_{\langle ab \rangle}$ and vorticity $\Delta_{[ab]}$ can be introduced by the splitting

$$a D_b \Delta_a = \frac{1}{3} \Delta h_{ab} + \Delta_{\langle ab \rangle} + \Delta_{[ab]}. \tag{A.29}$$

Taking into account the equations and constraints from the Bianchi identities, spatial inhomogeneities evolve according to the full non-linear equations (5) and (6).

As these equations are too complex to solve we seek to perturb the equations to first order. Therefore, we have to choose a background model, the FLRW metric, which reads in spherical coordinates

$$ds^2 = -dt^2 + a^2(t) \left[ \frac{d^2 r}{1 - K r^2} + r^2 d\Omega(\phi, \theta) \right]. \tag{A.30}$$

The curvature parameter $K$ can be $-1, 0, +1$. In this background the volume expansion is related to the Hubble parameter $H(t) := \dot{a}/a$ by $\Theta^{(0)}(t) = 3H(t)$ and thus Raychaudhuri's equation, the continuity equation and the Friedmann equation read

$$\dot{H} = -H^2 - \frac{\kappa}{6}(\rho + 3p) + \frac{1}{3}\Lambda, \quad \dot{\rho} = -3H(\rho + p), \quad \text{and} \tag{A.31}$$

$$H^2 = \frac{\kappa}{3}\rho - \frac{K}{a^2} + \frac{1}{3}\Lambda. \tag{A.32}$$

The evolution equation for linear density perturbations in a barotropic perfect fluid $p = \omega\rho$ in a FLRW universe $\Theta = 3H(t)$ with zero vorticity $\omega_{ab} = 0$ is then obtained from these equations setting the energy current density and the anisotropic stress to zero. This leads to [23]

$$\begin{aligned}
\ddot{\Delta} = &-2\left(1 - 3\omega + \frac{3}{2}c_s^2\right)H\dot{\Delta} \\
&+ \kappa\left[\left(\frac{1}{2} + 4\omega - 3c_s^2 - \frac{3}{2}\omega^2\right)\rho + (5\omega - 3c_s^2)\Lambda - \frac{12(\omega - c_s^2)K}{a^2}\right]\Delta \\
&+ c_s^2 D^2 \Delta.
\end{aligned} \tag{A.33}$$

Since for first order gauge invariant variables the zeroth order is zero we omit their perturbative labels and since appearing $\rho$'s, $p$'s and $\Theta$'s always occur together with a gauge invariant variable they must be of zeroth order such that we can also omit their superscripts.

It is useful to convert this equation into $k$-space by expanding $\Delta$ in scalar harmonics $\mathcal{Q}_k$ such that $\Delta = \int_k \Delta_k \mathcal{Q}_k$. The latter ones have the properties $\dot{\mathcal{Q}}_k = 0$ and $D^2 \mathcal{Q}_k = -\frac{k^2}{a^2}\mathcal{Q}_k$ (see appendix B for more information). In a spatially flat spacetime $K = 0$ the scalar harmonics

are plane waves. For a radiation dominated, flat universe we have $\omega = c_s^2 = 1/3$, $H = 1/(2t)$, $a(t) = a_0\sqrt{t/t_0}$, $\kappa\rho \equiv \rho^{(0)} = 3/(4t^2)$ and $K = 0$ such that in a comoving frame the dynamical equation for the $k$-th density perturbation mode yields

$$\frac{d^2\Delta_k}{dt^2} + \frac{1}{2t}\frac{d\Delta_k}{dt} - \frac{1}{2t^2}\left[1 - \frac{1}{6}\left(\frac{k}{a(t)H(t)}\right)^2\right]\Delta_k = 0, \tag{A.34}$$

which during radiation domination yields an oscillatory solution on sub-horizon scales and a linearly growing solution on super-horizon scales. During matter domination all modes grow with $\sim t^{2/3}$.

## A.5 Linear metric perturbations: Gravitational waves

In the $1+3$ covariant approach long range gravity effects are incorporated by the Weyl tensor and hence GWs are monitored by means of the transverse and tracefree components of its electric and magnetic parts. Linearizing the propagation equations Eq. (A.16), Eq. (A.17) and the constraints Eq. (A.18), Eq. (A.19) these equations read [87]

$$\dot{E}_{ab} = -\Theta E_{ab} + \operatorname{curl} H_{ab} - \frac{1}{2}\kappa\left[(\rho+p)\sigma_{ab} - D_{\langle a}q_{b\rangle} + \dot{\pi}_{ab} + \frac{1}{3}\Theta\pi_{ab}\right], \tag{A.35}$$

$$\dot{H}_{ab} = -\Theta H_{ab} - \operatorname{curl} E_{ab} - \frac{1}{2}\kappa\pi_{ab}, \tag{A.36}$$

$$D^b E_{ab} = \kappa\left(\frac{1}{3}\Theta q_b + \frac{1}{3}D_a\rho + \frac{1}{2}D^a\pi_{ab}\right) \quad \text{and} \quad D^b H_{ab} = \frac{1}{2}\kappa\left[2(\rho+p)\omega_a + \operatorname{curl} q_b\right]. \tag{A.37}$$

Hence in the absence of vorticity, the electric and magnetic parts of the Weyl tensor that are not sourced by density gradients ($D_a\rho = D_a\Theta = 0$) are transverse tensors for a perfect fluid ($\pi_{ab} = q_a = 0$) on a FLRW background ($\Theta = 3H(t)$) and due to Eq. (A.25) this is also true for the shear

$$D^a E_{ab} = 0, \quad D^a H_{ab} = 0 \quad \text{and} \quad D^a\sigma_{ab} = 0. \tag{A.38}$$

Using the linearized equations of motion for the shear Eq. (A.23) and the Weyl tensors, the latter ones can be eliminated from the discussion, see [29], to give the propagation equation

$$\ddot{\sigma}_{ab} + 5H(t)\dot{\sigma}_{ab} + \frac{1}{2}\kappa\rho(1-3\omega)\sigma_{ab} - D^2\sigma_{ab} = 0, \tag{A.39}$$

in the absence of curvature $K = 0$.

## A.6 Connection to Bardeen-formalism and Newtonian theory

The standard formalism frequently used for studying structure formation is based on the approach introduced by Bardeen [88]. In reference [52] Bruni et al. gave the transformation equations between the $1+3$-formalism presented here, and the $3+1$-slicing used by Bardeen. For later use and to connect to the more common formalism of Bardeen let us briefly repeat here the transformation rules. Primes denote in the following the conformal derivative $d\eta = dt/a$. We introduce the perturbed metric parametrized as

$$ds^2 = a^2(\eta)\left\{-(1+2A)d\eta^2 - 2B_\alpha d\eta dx^\alpha + [(1-2D)\delta_{\alpha\beta} + 2E_{\alpha\beta}]dx^\alpha dx^\beta\right\}, \tag{A.40}$$

$$\text{or} \quad g_{ab} = \eta_{ab} + \delta g_{ab} = a^2\left[\eta_{ab} + \begin{pmatrix} -2A & -B_\alpha \\ -B_\alpha & -2D\delta_{\alpha\beta} + 2E_{\alpha\beta}, \end{pmatrix}\right],$$

where $D = 1/6\cdot\delta g_a^a$ and $\delta^{\alpha\beta}E_{\alpha\beta} = 0$. The vector $\mathbf{B}$ is commonly split into a curl free, longitudinal part $\nabla\times\mathbf{B}^{\|} = 0$ and a source free, transverse $\nabla\cdot\mathbf{B}^{\perp} = 0$ such that $\mathbf{B} = \mathbf{B}^{\|} + \mathbf{B}^{\perp}$. While the

first part can be written in terms of a scalar potential $B$ the latter one originates from a vector potential. Similarly the tensor $E$ can be split into the components $E_{\alpha\beta} = E_{\alpha\beta}^{\parallel} + E_{\alpha\beta}^{\perp} + E_{\alpha\beta}^{T}$, where the first two again can be derived from a scalar and a vector potential $E$ and $\mathbf{E}$, respectively, and the last one fulfils the transversity conditions for tensors

$$E_{\alpha\beta}^{\parallel} = (\partial_\alpha \partial_\beta - \frac{1}{3}\delta_{\alpha\beta}\nabla^2)E, \tag{A.41}$$

$$E_{\alpha\beta}^{\perp} = -\frac{1}{2}(\partial_\beta E_\alpha + \partial_\alpha E_\beta) \quad \text{with} \quad \nabla \cdot \mathbf{E} = 0, \tag{A.42}$$

$$\delta^{\alpha\gamma}\partial_\gamma E_{\alpha\beta}^{T} = 0 \quad \text{and} \quad \delta^{\alpha\beta}E_{\alpha\beta}^{T} = 0. \tag{A.43}$$

The invariant form of the density variation $\delta := \delta\rho/\rho$ under a gauge transformation $x^a \to x^a + \epsilon\xi^a$ reads

$$\tilde{\delta} = \delta + 3\frac{a'}{a}(1+\omega)\xi^0. \tag{A.44}$$

With the splitting introduced above and the scalar potential $\mathbf{B}^{\parallel} =: -\nabla B$ the gauge invariant form of the scalar perturbations in terms of the metric perturbation parameters is

$$\tilde{\delta} = \delta - 3(1+\omega)\frac{a'}{a}(v - B), \tag{A.45}$$

where the fluid velocity perturbation is $\delta u_\alpha =: 1/a \cdot v_\alpha$, which also splits like the vector $\mathbf{B}$ in $\mathbf{v} = \mathbf{v}^{\parallel} + \mathbf{v}^{\perp}$ with the potential $\mathbf{v}^{\parallel} = -\nabla v$. A gauge is specified by choosing values for $v$ and $B$. Similarly, the other perturbative quantities can be made gauge invariant. The projected comoving density gradient $\Delta_a$ is

$$\mathbf{\Delta} = a\nabla\tilde{\delta} - 3a'(1+\omega)\mathbf{V}_c, \tag{A.46}$$

where $\mathbf{V}_c := \mathbf{v}^{\perp} - \mathbf{B}^{\perp}$. Hence its divergence yields

$$\Delta = \nabla^2\tilde{\delta}, \tag{A.47}$$

due to $\nabla \cdot \mathbf{v}^{\perp} = \nabla \cdot \mathbf{B}^{\perp} = 0$. Eq. (A.47) is the desired connection between the common gauge invariant Bardeen variable for the density perturbation and the $1+3$ scalar variations.

The shear tensor $\sigma_{\alpha\beta}$, describing GWs in the $1+3$ formalism, is closely related to the transverse and tracefree part of the tensor perturbation $E_{\alpha\beta}^{\perp}$ which equals the commonly used $h_{\alpha\beta}$ in the transverse tracefree gauge and is gauge invariant by itself. The shear tensor expressed in terms of Bardeen parametrized metric perturbation reads

$$\sigma_{\alpha\beta} = a(\nabla_{\alpha\beta}V_S + \nabla_{(\alpha}V_{S\,\beta)} + E_{\alpha\beta}^{T\prime}), \tag{A.48}$$

where $\Delta_{\alpha\beta} := \nabla_\alpha\nabla_\beta - \frac{1}{3}\delta_{\alpha\beta}\nabla^2$ and $V_S := v - D'$. For the purpose of this work we will only need the relations between the projected density gradient and the shear with Bardeens variables given in Eqs. (A.47) and (A.48), respectively. Analog expressions for other quantities can be found in [52].

## A.7 Important identities in 1+3 covariant theory

The orthogonal projected gradient and the time derivative of the orthogonal projection operator $h_{ab}$ meet the relations [28, 52]

$$\mathrm{D}_a h_{bc} = 0, \tag{A.49}$$

$$\mathrm{D}^a h_{ab} = u_b \Theta, \tag{A.50}$$

$$\dot{h}_{ab} = u_b A_a + u_a A_b. \tag{A.51}$$

Calculating the projected gradient of the four velocity gives

$$D_b u_a = \sigma_{ab} + \omega_{ab} + \frac{1}{3}\Theta h_{ab}. \tag{A.52}$$

Also important are the commutation laws for the derivatives which we simply repeat from reference [28]. For a scalar $f$, a vector $v_a$ and a tensor $S_{ab}$ we have for the spatial derivative

$$D_{[a}D_{b]}f = -\omega_{ab}\dot{f}, \tag{A.53}$$

$$D_{[a}D_{b]}v_c = -\omega_{ab}\dot{v}_{\langle a\rangle} + \frac{1}{2}\mathcal{R}_{dcba}v^d, \tag{A.54}$$

$$D_{[a}D_{b]}S_{cd} = -\omega_{ab}h_c{}^e h_d{}^f \dot{S}_{ef} + \frac{1}{2}(\mathcal{R}_{ecba}S^e{}_d + \mathcal{R}_{edba}S_c{}^e), \tag{A.55}$$

where $\mathcal{R}_{abcd}$ is the Riemann tensor in the local rest space of the observer.
Similarly the time derivative and the space derivative do not commute in general

$$D_a\dot{f} - h_a{}^b(D_b f)^{\cdot} = -\dot{f}A_a + \frac{1}{3}\Theta D_a f + D_b f\left(\sigma^b{}_a + \omega^b{}_a\right). \tag{A.56}$$

## B  Harmonic Decomposition

It is convenient to expand all scalars, vectors and tensors in harmonic functions. No matter if scalar harmonic $\mathcal{Q}_k$, vector harmonic $\mathcal{Q}_{k,a}$ or tensor harmonic $\mathcal{Q}_{k,ab}$, their defining property is to be an eigenfunction of the orthogonal projected Laplace operator (Laplace-Beltrami equation)

$$D^2\mathcal{Q}_{k,\{\ ,a,ab\}} = -\frac{k^2}{a^2}\mathcal{Q}_{k,\{\ ,a,ab\}}, \tag{B.1}$$

with eigenvalue $-k^2/a^2$. In case of a flat space $K = 0$ the orthogonal projected Laplace operator $D^2$ reduces to the usual Laplace operator $\nabla^2/a^2$ such that the harmonic functions $\mathcal{Q}$ are Fourier transforms [80, 89]. Scalar, vector and tensor modes thus transform like

$$f(\mathbf{x},t) = \int d\mathbf{k} f_{\mathbf{k}}(t)\mathcal{Q}_{\mathbf{k}}, \qquad\qquad V_a^\perp(\mathbf{x},t) = \int d\mathbf{k}\sum_{m=-1,1} V_{\mathbf{k}}^{\perp[m]}\mathcal{Q}_{\mathbf{k},a}^{[m]}, \tag{B.2}$$

$$S_{ab}^T(\mathbf{x},t) = \int d\mathbf{k}\sum_{m=-2,2} S_{\mathbf{k}}^{T[m]}\mathcal{Q}_{\mathbf{k},ab}^{[m]}, \tag{B.3}$$

with

$$\mathcal{Q}_{\mathbf{k}} = \exp(i\mathbf{k}\cdot\mathbf{x}),$$

$$\mathcal{Q}_{\mathbf{k},a}^{[\pm1]} = \frac{-i}{\sqrt{2}}(\mathbf{e}_1 \pm i\mathbf{e}_2)_a \exp(i\mathbf{k}\cdot\mathbf{x}),$$

$$\mathcal{Q}_{\mathbf{k},ab}^{[\pm2]} = -\sqrt{\frac{3}{8}}(\mathbf{e}_1 \pm i\mathbf{e}_2)_a(\mathbf{e}_1 \pm i\mathbf{e}_2)_b \exp(i\mathbf{k}\cdot\mathbf{x}),$$

where $\mathbf{e}_1$ and $\mathbf{e}_2$ are orthonormal basis vectors. The scalar part of a vector and the scalar- and vector part of a tensor expand like

$$V_a^{\parallel} = \int d\mathbf{k}\left(-i\frac{k_a}{k}\right)V_{\mathbf{k}}\mathcal{Q}_{\mathbf{k}}, \tag{B.4}$$

$$S_{ab}^{\parallel} = \int d\mathbf{k}\left(-\frac{k_a k_b}{k^2} + \frac{1}{3}h_{ab}\right)S_{\mathbf{k}}\mathcal{Q}_{\mathbf{k}}, \tag{B.5}$$

$$S_{ab}^{\perp} = \int d\mathbf{k}\sum_{m=-1,+1}\left(-\frac{i}{2k}\right)(k_a S_{\mathbf{k}}^{\perp[m]}\mathcal{Q}_{\mathbf{k},b}^{[m]} + k_b S_{\mathbf{k}}^{\perp[m]}\mathcal{Q}_{\mathbf{k},a}^{[m]}), \tag{B.6}$$

respectively.

# C   Decay of the source after the FoPT

We will now show that the GW source decays sufficiently fast after the PT such that we can take the right hand side of Eq. (82) to zero. We first perform the adimensional change of variables

$$\tilde{\delta}''(\kappa,\tau) + \frac{a[\tau]'}{a[\tau]}\tilde{\delta}'(\kappa,\tau) + \frac{1}{3}\frac{a_*^2}{a[\tau]^2}\kappa^2\tilde{\delta}(\kappa,\tau) = 8\frac{H[\tau]^2}{H_*^2}\frac{a_*^4}{a[\tau]^4}\cdot\Omega_{\text{GW}}(\kappa,\tau). \tag{C.1}$$

Note that $\Omega_{\text{GW}}(\kappa,\tau > \tau_f) = \Omega_{\text{GW}}(\kappa,\tau_f) := \Omega_{\text{GW}}(\kappa)$ is a constant in time, where $\tau_f = \tau_* + 1/r_\beta$ is the time when the PT ends. We can now solve the homogeneous part of Eq. (C.1) for a general $a[\tau]$:

$$\tilde{\delta}_h(\kappa,\tau > \tau_f) = C_\kappa \cos\left[\frac{a_*\kappa}{\sqrt{3}}\int_{\tau_f}^{\tau}\frac{1}{a[\tau']}d\tau'\right] + D_\kappa \sin\left[\frac{a_*\kappa}{\sqrt{3}}\int_{\tau_f}^{\tau}\frac{1}{a[\tau']}d\tau'\right], \tag{C.2}$$

which in the radiation dominated universe simplifies to $a(\tau) = a_*\sqrt{\frac{\tau}{\tau_*}}$ and $H(\tau) = H_*\frac{1}{2\tau}$

$$\tilde{\delta}_h(\kappa,\tau > \tau_f) = C_\kappa \cos\left[\frac{2\tau_*\kappa}{\sqrt{3}}\left(\sqrt{\tau} - \sqrt{\tau_f}\right)\right] + D_\kappa \sin\left[\frac{2\tau_*\kappa}{\sqrt{3}}\left(\sqrt{\tau} - \sqrt{\tau_f}\right)\right]. \tag{C.3}$$

This is the solution sourced solely by the GW energy during the PT. Turning now to the solution sourced by the GW after the PT, using the variation of parameters method we find

$$\tilde{\delta}(\kappa,\tau > \tau_f) = \tilde{\delta}_h(\kappa,\tau > \tau_f) + \tilde{\delta}_{nh}(\kappa,\tau > \tau_f), \tag{C.4}$$

$$\tilde{\delta}_{nh}(\kappa,\tau > \tau_f) = C_\Omega(\kappa,\tau)\cos\left[\frac{2\sqrt{\tau_*}\kappa}{\sqrt{3}}\left(\sqrt{\tau} - \sqrt{\tau_f}\right)\right]$$
$$+ D_\Omega(\kappa,\tau)\sin\left[\frac{2\sqrt{\tau_*}\kappa}{\sqrt{3}}\left(\sqrt{\tau} - \sqrt{\tau_f}\right)\right], \tag{C.5}$$

$$C_\Omega(\kappa,\tau) = -\int_{\tau_f}^{\tau}\frac{\sqrt{3}8\frac{H[\tau']^2}{H_*^2}\frac{a_*^4}{a[\tau']^4}\cdot\Omega_{\text{GW}}(\kappa)\sqrt{\frac{\tau'}{\tau_*}}\sin\left(\frac{2\kappa\sqrt{\tau_*}\left(\sqrt{\tau'} - \sqrt{\tau_f}\right)}{\sqrt{3}}\right)}{\kappa}d\tau', \tag{C.6}$$

$$D_\Omega(\kappa,\tau) = \int_{\tau_f}^{\tau}\frac{\sqrt{3}8\frac{H[\tau']^2}{H_*^2}\frac{a_*^4}{a[\tau']^4}\cdot\Omega_{\text{GW}}(\kappa)\sqrt{\frac{\tau'}{\tau_*}}\cos\left(\frac{2\kappa\sqrt{\tau_*}\left(\sqrt{\tau'} - \sqrt{\tau_f}\right)}{\sqrt{3}}\right)}{\kappa}d\tau'. \tag{C.7}$$

For demonstration purposes we will now recombine the trigonometric functions into a single one with a phase by using the identity

$$C \cos x + D \sin x = A \sin(x + \phi), \tag{C.8}$$

where the new amplitude is given by $A = \sqrt{C^2 + D^2}$ and the relative phase $\phi = \arctan \frac{C}{D}$. Applying this identity into Eqs. (C.3) and (C.5) we obtain

$$\tilde{\delta}_h(\kappa, \tau > \tau_f) = A_\kappa \sin\left[\frac{2\tau_* \kappa}{\sqrt{3}}\left(\sqrt{\tau} - \sqrt{\tau_f}\right) + \phi_\kappa\right], \tag{C.9}$$

$$\tilde{\delta}_{nh}(\kappa, \tau > \tau_f) = A_\Omega(\kappa, \tau) \sin\left[\frac{2\sqrt{\tau_*}\kappa}{\sqrt{3}}\left(\sqrt{\tau} - \sqrt{\tau_f}\right) + \phi_\Omega\right]. \tag{C.10}$$

We now compare the relative sizes of the amplitudes $A_\kappa$, the amplitude of the solution sourced by the gravitational wave energy at $t_f$, and $A_\Omega(\kappa, \tau_{eq})$, the amplitude of the solution sourced by the gravitational wave energy after $t_f$ at $t_{eq}$. As can be seen in Fig. (14), the homogeneous solution dominates and thus taking the right hand side of Eq. (82) to zero is a sound approximation.

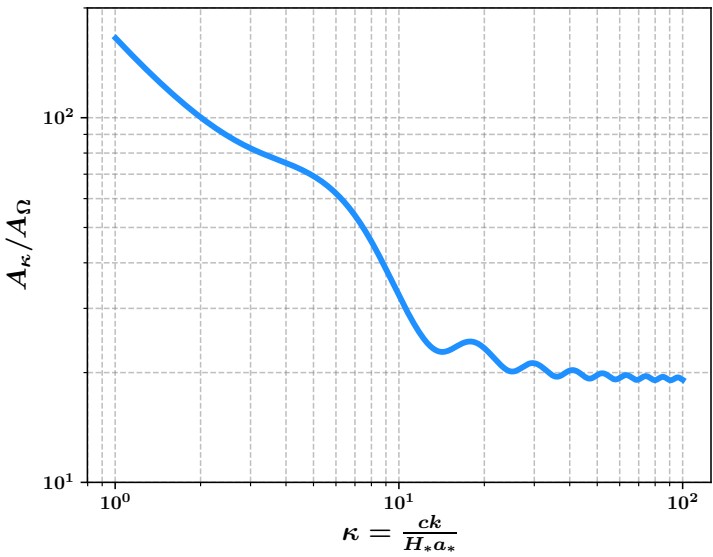

Figure 14: Ratio between the amplitudes in Eqs. (C.9) and (C.10) evaluated at $t_{eq}$ for a benchamark strong PT with $\alpha \to \infty$, $r_\beta = 1$ and $t_* = 10^{10}$ s. Note that the ratio is always bigger than 20, meaning that the solution sourced by the decaying GW source after $t_f$ can be safely neglected in Eq. (82).

# D  Analytical solution of the GW sourced wave equation in the small and high wave number limit

We start from the source given by Eq. (64). By performing the variable tranformations $t = H_* \tau$ and $k = \frac{H_* a_*}{c} \kappa$ and by slightly abusing the notation, we obtain

$$L(\tau) = \frac{4\nu\beta}{H_*^2}(\tau - \tau_*)(\tau_f - \tau), \tag{D.1}$$

$$f^2(\kappa, \tau) = L(\tau)^2 \frac{\nu\epsilon}{\beta} \frac{1 + \left(\frac{\kappa}{3}L(\tau)\right)^2}{1 + \left(\frac{\kappa}{3}L(\tau)\right)^2 + \left(\frac{\kappa}{3}L(\tau)\right)^6}, \tag{D.2}$$

$$\Delta(\kappa, \tau) = \kappa^3 \left(r_\beta\right)^2 (\tau - \tau_*)^2 f^2\left(\kappa, \frac{\tau + \tau_*}{2}\right), \tag{D.3}$$

$$\Omega_{\mathrm{GW}}^{\log}\left(\kappa', \tau\right) = \left(r_\beta\right)^{-2} \frac{\alpha^2}{(1+\alpha)^2} \Delta(\kappa, \tau), \tag{D.4}$$

$$\Omega_{\mathrm{GW}}(\kappa, \tau) = \int_{\kappa_{\min}}^{\kappa} \Omega_{\mathrm{GW}}^{\log}\left(\kappa', \tau\right) \mathrm{d}\log\kappa'. \tag{D.5}$$

Then, we want to expand the quantity $\Omega_{\mathrm{GW}}(\kappa, \tau)$ around $\kappa = 0$ and $\kappa \to \infty$. The expansion around $\kappa = 0$ can simply be done by first expanding $\Omega_{\mathrm{GW}}^{\log}$ and then integrating:

$$\Omega_{\mathrm{GW}}^{\mathrm{low}} = C_{\mathrm{GW}}^{\mathrm{low}}(\kappa - \kappa_{\min})^3(\tau_* - \tau)^4(\tau + \tau_* - 2\tau_f)^2, \quad \text{with} \tag{D.6}$$

$$C_{\mathrm{GW}}^{\mathrm{low}} = \frac{1}{3}r_\beta \frac{\alpha^2}{(1+\alpha)^2}\epsilon. \tag{D.7}$$

Note, however, that the same cannot be applied for the high $\kappa$ limit, since the integration of $\Omega_{\mathrm{GW}}^{\mathrm{low}}$ will necessarily run over small values of $\kappa'$. In order to solve this issue, we first obtain the value of $\kappa_{\mathrm{peak}}$ defined as

$$\left(\Omega_{\mathrm{GW}}^{\log}(\kappa_{\mathrm{peak}}, \tau_f)\right)^{\mathrm{low}} = \left(\Omega_{\mathrm{GW}}^{\log}(\kappa_{\mathrm{peak}}, \tau_f)\right)^{\mathrm{high}} \to \kappa_{\mathrm{cross}} = 3\frac{H_*}{\beta}\frac{c}{\nu}\frac{1}{(\tau_f - \tau_*)^2}, \tag{D.8}$$

which is the value of $\kappa$ for which the low and high $\kappa$ limits of the GW energy density intersect at $\tau = \tau_f$. Then for the high $\kappa$ approximation of $\Omega_{\mathrm{GW}}$ we can write

$$\begin{aligned}
\Omega_{\mathrm{GW}}^{\mathrm{high}} &= \int_{\kappa_{\min}}^{\kappa_{\mathrm{peak}}} \left(\Omega_{\mathrm{GW}}^{\log}(\kappa, \tau)\right)^{\mathrm{low}} \mathrm{d}\log\kappa' + \int_{\kappa_{\mathrm{peak}}}^{\kappa} \left(\Omega_{\mathrm{GW}}^{\log}(\kappa, \tau)\right)^{\mathrm{high}} \mathrm{d}\log\kappa' \\
&= C_{\mathrm{GW}}^{\mathrm{low}}(\kappa_{\mathrm{peak}} - \kappa_{\min})^3(\tau_* - \tau)^4(\tau + \tau_* - 2\tau_f)^2 + C_{\mathrm{GW}}^{\mathrm{high}}\frac{\kappa - \kappa_{\min}}{\kappa\kappa_{\min}}\frac{1}{(\tau + \tau_* - 2\tau_f)^2},
\end{aligned} \tag{D.9}$$

and $C_{\mathrm{GW}}^{\mathrm{high}} = 81\frac{c}{\nu}\frac{H_*^3}{\beta^3}\frac{\alpha^2}{(1+\alpha)^2}\epsilon\kappa_{\mathrm{eff}}$.

In these two regimes the differential equation Eq. (84) can be solved analytically. Note, that the equation is only valid during the FPT, i.e. $\tau_* < \tau < \tau_f$. The homogeneus part of the equation is given by $\tilde{\delta}''(\kappa, \tau) + \frac{1}{3}\kappa^2\tilde{\delta}(\kappa, \tau) = 0$ with a trivial solution: $\tilde{\delta}_h(\kappa, \tau) = A_\kappa \cos\left(\frac{\kappa}{\sqrt{3}}\tau\right) + B_\kappa \sin\left(\frac{\kappa}{\sqrt{3}}\tau\right)$, where $A_\kappa$ and $B_\kappa$ are given by the initial conditions. We can then use the variation of parameters method to obtain the solution to the non-homogeneous solution which is given by

$$\begin{aligned}
\delta^{(2)}(\kappa, \tau) = \delta_h^{(2)}(\kappa, \tau) + \frac{8\sqrt{3}}{\kappa}\Bigg[&\sin\left(\frac{\kappa}{\sqrt{3}}\tau\right)\int_{\tau_*}^{\tau_f}\cos\left(\frac{\kappa}{\sqrt{3}}\tilde{\tau}\right)\Omega_{\mathrm{GW}}(\tilde{\tau}, \kappa)\mathrm{d}\tilde{\tau} \\
&-\cos\left(\frac{\kappa}{\sqrt{3}}\tau\right)\int_{\tau_*}^{\tau_f}\sin\left(\frac{\kappa}{\sqrt{3}}\tilde{\tau}\right)\Omega_{\mathrm{GW}}(\tilde{\tau}, \kappa)\mathrm{d}\tilde{\tau}\Bigg]. \tag{D.10}
\end{aligned}$$

Since $\Omega_{GW}(\tau,\kappa)$ is 0 for times before the phase transition the non-homogeneus part is 0 when $\tau < \tau_*$. The lower limit of the integral becomes $\tau_*$ if $\tau > \tau_*$. If we assume that the source decays quickly after $\tau_f$ the upper limit of the integral becomes $\tau_f$ if $\tau > t_f$. Therefore, the expression evaluated at $\tau > \tau_f$, i.e. right after the end of the phase transition, becomes

$$
\tilde{\delta}(\kappa,\tau > \tau_f) = \tilde{\delta}_h(\kappa,\tau) + \frac{8\sqrt{3}}{\kappa}\left[\sin\left(\frac{\kappa}{\sqrt{3}}\tau\right)\int_{\tau_*}^{\tau_f}\cos\left(\frac{\kappa}{\sqrt{3}}\tilde{\tau}\right)\Omega_{GW}(\tilde{\tau},\kappa)d\tilde{\tau}\right.
$$
$$
\left. -\cos\left(\frac{\kappa}{\sqrt{3}}\tau\right)\int_{\tau_*}^{\tau_f}\sin\left(\frac{\kappa}{\sqrt{3}}\tilde{\tau}\right)\Omega_{GW}(\tilde{\tau},\kappa)d\tilde{\tau}\right]. \tag{D.11}
$$

We can see that before $\tau_*$, $\tilde{\delta}$ behaves as an harmonic oscillating function with constants $A_k$ and $B_k$ given by an initial value. During the phase transition, the time dependence of $\tilde{\delta}$ will be a complicated function. However, for times $\tau > \tau_f$ the integrals become constants in time (they still depend on $\kappa$) and therefore $\tilde{\delta}$ is again an harmonic oscillation with modified amplitudes for each $\kappa$.

We can now solve these integrals in the limits for low and high $\kappa$ by substituting $\Omega_{GW}$ in Eq. (D.11) by Eqs. (D.7) and (D.9). The solution for $\tilde{\delta}^{(2)\text{low}}$ becomes

$$
\tilde{\delta}^{\text{low}}(\kappa,\tau > \tau_f) = \tilde{\delta}_h(\kappa,\tau) + \tilde{\delta}_{GW}(\kappa,\tau)
$$
$$
= \left(A_\kappa + A_{GW\kappa}^{\text{low}}\right)\cos\left(\frac{\kappa\tau}{\sqrt{3}}\right) + \left(B_\kappa + B_{GW\kappa}^{\text{low}}\right)\sin\left(\frac{\kappa\tau}{\sqrt{3}}\right), \tag{D.12}
$$

where $A_{GW\kappa}^{\text{low}}$ and $B_{GW\kappa}^{\text{low}}$ are the 'modified' amplitudes due to the effect of the FPT in the low k limit. They are given by

$$
A_{GW\kappa}^{\text{low}} = -\frac{24C_{GW}^{\text{low}}\left(\kappa^3 - \kappa_{\min}^3\right)}{\kappa^8}
$$
$$
\left[2\sqrt{3}r_\beta^{-1}\kappa\left(2160\sin\left(\frac{\kappa\tau_*}{\sqrt{3}}\right) + \left(1080 + 36r_\beta^{-2}\kappa^2 + r_\beta^{-4}\kappa^4\right)\sin\left(\frac{\kappa\tau_f}{\sqrt{3}}\right)\right)\right.
$$
$$
+ \left(19440 + 216r_\beta^{-2}\kappa^2 - 6r_\beta^{-4}\kappa^4 - r_\beta^{-6}\kappa^6\right)\cos\left(\frac{\kappa\tau_f}{\sqrt{3}}\right)
$$
$$
\left. + 432\left(2r_\beta^{-2}\kappa^2 - 45\right)\cos\left(\frac{\kappa\tau_*}{\sqrt{3}}\right)\right], \tag{D.13}
$$
$$
B_{GW\kappa^{\text{low}}} = \frac{24C_{GW}^{\text{low}}\left(\kappa^3 - \kappa_{\min}^3\right)}{\kappa^8}
$$
$$
\left[2\sqrt{3}r_\beta^{-1}\left(2160\kappa\cos\left(\frac{\kappa\tau_*}{\sqrt{3}}\right) + \kappa\left(1080 + 36r_\beta^{-2}\kappa^2 + r_\beta^{-4}\right)\cos\left(\frac{\kappa\tau_f}{\sqrt{3}}\right)\right)\right.
$$
$$
+ \left(-19440 - 216r_\beta^{-2}\kappa^2 + 6r_\beta^{-4}\kappa^4 + r_\beta^{-6}\kappa^6\right)\sin\left(\frac{\kappa\tau_f}{\sqrt{3}}\right)
$$
$$
\left. + 432\left(45 - 2r_\beta^{-2}\kappa^2\right)\sin\left(\frac{\kappa\tau_*}{\sqrt{3}}\right)\right]. \tag{D.14}
$$

Analogously, for high $\kappa$ we have

$$
\tilde{\delta}^{\text{high}}(\kappa,\tau > \tau_f) = \tilde{\delta}_h(\kappa,\tau) + \tilde{\delta}_{GW}^{\text{high}}(\kappa,\tau)
$$
$$
= \left(A_\kappa + A_{GW\kappa}^{\text{high}}\right)\cos\left(\frac{\kappa\tau}{\sqrt{3}}\right) + \left(B_\kappa + B_{GW\kappa}^{\text{high}}\right)\sin\left(\frac{\kappa\tau}{\sqrt{3}}\right), \tag{D.15}
$$

with the constants $A_{GW\kappa}^{high}$ and $B_{GW\kappa}^{high}$ given by

$$
\begin{aligned}
A_{GW}^{high} = A_{GW\kappa_{peak}}^{low} &- \frac{4C_{GW}^{high}(\kappa - \kappa_{peak})}{3\sqrt{3}\kappa^2\kappa_{peak}}\Bigg[6r_\beta \sin\left(\frac{\kappa\tau_f}{\sqrt{3}}\right) - 3r_\beta \sin\left(\frac{\kappa\tau_*}{\sqrt{3}}\right) \\
&+ 2\sqrt{3}\kappa\left(\left(\text{Ci}\left(\frac{\kappa}{\sqrt{3}r_\beta}\right) - \text{Ci}\left(\frac{2\kappa}{\sqrt{3}r_\beta}\right)\right)\cos\left(\frac{\kappa(2\tau_f - \tau_*)}{\sqrt{3}}\right)\right. \\
&+ \left.\left(\text{Si}\left(\frac{\kappa}{\sqrt{3}r_\beta}\right) - \text{Si}\left(\frac{2\kappa}{\sqrt{3}r_\beta}\right)\right)\sin\left(\frac{\kappa(2\tau_f - \tau_*)}{\sqrt{3}}\right)\right)\Bigg],
\end{aligned}
\tag{D.16}
$$

$$
\begin{aligned}
B_{GW}^{high} = B_{GW\kappa_{peak}}^{low} &- \frac{4C_{GW}^{high}(\kappa - \kappa_{peak})}{3\sqrt{3}\kappa^2\kappa_{peak}}\Bigg[6r_\beta \cos\left(\frac{\kappa\tau_f}{\sqrt{3}}\right) - 3r_\beta \cos\left(\frac{\kappa\tau_*}{\sqrt{3}}\right) \\
&+ 2\sqrt{3}\kappa\left(\left(\text{Ci}\left(\frac{2\kappa}{\sqrt{3}r_\beta}\right) - \text{Ci}\left(\frac{\kappa}{\sqrt{3}r_\beta}\right)\right)\sin\left(\frac{\kappa(2\tau_f - \tau_*)}{\sqrt{3}}\right)\right. \\
&+ \left.\left(\text{Si}\left(\frac{\kappa}{\sqrt{3}r_\beta}\right) - \text{Si}\left(\frac{2\kappa}{\sqrt{3}r_\beta}\right)\right)\cos\left(\frac{\kappa(2\tau_f - \tau_*)}{\sqrt{3}}\right)\right)\Bigg].
\end{aligned}
\tag{D.17}
$$

Here the functions $\text{Ci}(x)$ and $\text{Si}(x)$ are the CosIntegral and SinIntegral functions, respectively, defined by $\text{Ci}(x) = \int_0^x \frac{\cos t}{t}dt$ and $\text{Si}(x) = \int_0^x \frac{\sin t}{t}dt$.

In order to compare these results with the numerical solution, we evaluate Eqs. (D.12) and (D.15) in $\tau = \tau_f$. As a a benchmark scenario, we choose extreme values for the phase transition parameters: $\beta = H_*$, $\alpha \to \infty$, $v = c$, $k_{eff} = 1$, $\epsilon = 0.01$, which yields $\kappa_{peak} = 3$. In Fig. 15 we show the analytic results in the two $\kappa$-regimes compared to the numerical solution for zero initial conditions.

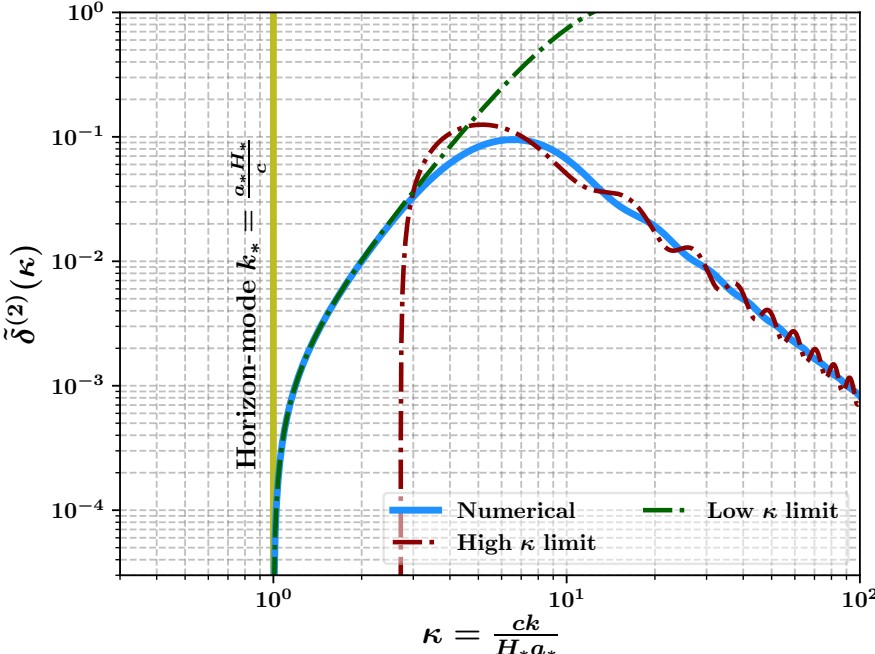

Figure 15: Comparison between the analytic solution for the density perturbations at second order $\tilde{\delta}^{(2)}$ evaluated at $\tau = \tau_f$ in the low- and high $\kappa$ limit with the full numerical solution.

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
