# Peer review of "Gravitational wave induced baryon acoustic oscillations"

_SciPost Physics, doi:SciPost Phys. 12, 114 (2022)_

## Round 1 · Referee Report · Anonymous (Referee 1) · 2022-2-15

Strengths

The paper is interesting and discussed a novel physical phenomena

Weaknesses

The observational signature is very weak and may not ever be truly observed.

Report

The authors satisfactorily replied to my previous report.
I now recommend the paper for publication.

---

## Round 1 · List of Changes

Response to the Referee

(i) Response to: “Give an estimate of the ∆σ term at late times.”

Response: There is a rough estimate of the ∆σ terms at the time of the phase transition on
p. 26 of the previous version of the manuscript. For later times, the shear amplitude will decay
further due to cosmic expansion. However, in order to give more visibility to this estimation we
have created Eq. (74) on p. 19 of the revised version rather than keeping it in a text block.

(ii) Response to: “Most of section 2 is standard and can be found in text books. I recommend to
put this section into an appendix and streamline the main text.”

Response: Following the referee’s suggestion we have moved much of section 2 to the appendix
and streamlined the text accordingly.

(iii) Response to: “In section 3, assumption 2 you neglect anisotropic stresses also at second order.
Please comment why you may neglect the term ρ v_a v_b which would contribute an isotropic stress.”

Response: The term ρ^(0) v_a v_b does indeed not enter explicitly into the calculation. It emerges
from the decomposition of the energy momentum tensor T_ab = ρ u_a u_b + . . . . This part of the energy momentum
tensor enters then the non-linear equations for (d/dt∆)_<a> via the projection of the Einstein field equations with the
velocity u^a such that R_ab u^a u^b = 0.5(ρ + 3p) − Λ (R_ab denotes the Ricci tensor).
Therefore, when expanding the non-linear equations for (d/dt∆)_</a><a> it automatically accounts for the
term ρ^{(0)} v_a v_b . Additionally, all velocity perturbations are indirectly included via the shear,
volume expansion, vorticity and acceleration which emerge from
the decomposition of the fundamental four-velocity ∇_b u_a . This can be also seen from Eqs. (80)
ff. in Ref. [1]. A non-uniform density in the stresses ρ v_a v_b would only enter at even higher order.

(iv) Response to: “κ_eff in Eq. (117) is not defined.”

Response: κ_eff is first mentioned on p. 14 of the revised version, where it is defined (“The
fraction of the released energy actually transmitted to the kinetic energy of the fluid is provided
by the efficiency factor κ_eff .”). However, we have added a small comment after Eq. (65) of the
revised version for clarity.

(v) Response to: “On p26 ’comoving derivative’, I think you mean the derivative w.r.t conformal
time.”

Response: The referee is correct and we have updated the manuscript accordingly on page 19
of the revised version.

(vi) Response to: “In the late time power spectrum which is observed in galaxy surveys, non-linearities
are important and these may very well ’wash out’ the small signal from GWs. This should at
least be briefly discussed.”

Response: We agree with the referee that non-linear structure growth will affect the calculated
signatures. However, these signatures will also serve as seeds for the non-linear evolution and
therefore contribute to the late time growth of structure. In this sense, we would rather say
that the signatures are overlaid but not washed out by non-linear evolution. We have added a
comment to the conclusions of the paper (p. 30), accordingly.

Additions:

(i) LIGO collaboration → LIGO-Virgo-KAGRA collaboration on p. 1.

(ii) Added Citation of arXiv [2012.11614]; citation number 11 of the new manuscript.

(iii) Formulas (D.1)-(D.5), (D.9), (D.13)-(D.14), (D.16)-(D.17) have been rearranged and be-
came more readable.

References

[1] Marco Bruni, Peter K.S. Dunsby, and George F.R. Ellis. Cosmological perturbations and the
physical meaning of gauge invariant variables. Astrophys. J., 395:34–53, 1992.</a>

---

## Editorial Decision

published